# Delineation of Potential Groundwater Zones and Assessment of Their Vulnerability to Pollution from Cemeteries Using GIS and AHP Approaches Based on the DRASTIC Index and Specific DRASTIC

Vanessa Gonçalves [1,2], Antonio Albuquerque [1,2,*], Pedro Gabriel Almeida [1,2], Luís Ferreira Gomes [1,2] and Victor Cavaleiro [1,2]

1  Department of Civil Engineering and Architecture, University of Beira Interior, Calcada Fonte do Lameiro 6, 6200-358 Covilha, Portugal; vanessa_alvane@hotmail.com (V.G.); galmeida@ubi.pt (P.G.A.); lmfg@ubi.pt (L.F.G.); victorc@ubi.pt (V.C.)
2  GeoBioTec, University of Beira Interior, Calcada Fonte do Lameiro 6, 6200-358 Covilha, Portugal
*  Correspondence: antonio.albuquerque@ubi.pt; Tel.: +351-275-329-734

**Abstract:** The risk of aquifer contamination is determined by the interaction between the pollutant load and the vulnerability of an aquifer. Owing to the decomposition of bodies and degradation of artefacts, cemeteries may have a negative impact on groundwater quality and suitability for use due to the leaching of organic compounds (e.g., biodegradable organics, pharmaceuticals, and formaldehyde), inorganic compounds (e.g., nitrate and heavy metals), pathogenic bacteria, and viruses. Factors such as burial and soil type, rainfall amount, and groundwater depth may increase aquifer vulnerability to pollutants generated in cemeteries. The potential for groundwater contamination was investigated in two cemeteries of the Soure region in Portugal (Samuel–UC9 and Vinha da Rainha–UC10), using the classic DRASTIC model, followed by some adjustments, depending on the particularities of the locations, resulting in a Final Classification considered as Specific DRASTIC. By combining Remote Sensing (RS), Geographic Information System (GIS), and Analytical Hierarchy Process (AHP), groundwater potential zones (GWPZs) were identified, and aquifer vulnerability was assessed, which included the elaboration of thematic maps using GIS operation tools. The maps allowed for the identification of areas with different susceptibilities to contamination: from "Low" to "Very high" for the DRASTIC index and from "Very Low" to "Very high" for the Specific DRASTIC index. Although the difference between the UC9 and UC10 cemeteries is negligible, UC10 is more vulnerable because of its proximity to the community and critically important mineral water resources (such as Bicanho Medical Spa). The Specific model seems better-suited for describing vulnerability to cemeteries. Although there is limited groundwater quality data for the area, the development of vulnerability maps can identify areas that can be sensitive spots for groundwater contamination and establish procedures for pollution prevention.

**Keywords:** cemeteries; groundwater contamination; vulnerability map; GIS tools; DRASTIC; Specific DRASTIC





## 1. Introduction

Cemeteries are places where institutional funeral practices take place and have a special meaning for storing and transforming dead bodies and serving as a collective historical memory [1–6].

People and societies have long considered contaminated groundwater near established and unplanned cemeteries to be an urgent concern [7], because it is a slow, chronic, and asymptomatic process [8,9] and should be referred to as decomposition labs [10]. Human cadavers typically contain approximately 35% organic material, 15% bone, and

50% water [11]. According to the authors [11], when a person weighing about 70 kg decomposes, 30–40 L of necro leachate is released into the environment between 72 h and 3 years after death [12] and it takes between 15 and 25 years for the person to completely break down into a skeleton [11]. The decomposition process releases organic materials, inorganic materials, gases, and trace elements into the groundwater, which harms both the environment and humans [9,12,13]. The primary sources of pollution in cemeteries, according to Guttman et al. [1], are materials used to manufacture coffins and embalming fluids. Toxic metals (e.g., Fe, Cu, Ni, Pb, and Zn) are released into the soil by varnishes, sealants, metal handles, and decorations found on wooden coffins [11,14–20].

As the world population increases and because of urban land development, cemeteries that were previously located on the outskirts of communities are now located in their centre. Many regions of the world have reported graveyard soil contaminated with extreme physical, chemical, and biological elements [21–23]. Certain dangerous substances can linger in the atmosphere for extended periods as ultrafine or nanoparticle-sized particles [24–26]. In the soils of urban cemeteries in Passo Fundo, Brazil, the concentrations of toxic metals were higher than those naturally occurring in control samples [2,27]. Dent's [28] research at the Australian Botanical Cemetery indicates that the electrical conductivity (or salinity) around recent burials has increased noticeably. High levels of $Cl^-$, $NO_3^-$, $NO_2^-$, $NH_4^+$, $PO_4^{3-}$, Fe, $Na^+$, $K^+$, and $Mg^{2+}$ were found beneath the cemetery [28]. Additionally, previous research has shown that a variety of contaminants, including bacteria, viruses, phosphorus, and nitrogen, can contaminate groundwater and pose a health risk to the general public [1,2,5,10,29–34].

Groundwater contamination is significantly influenced by burial practices, including individual or collective graves, grave depth, proximity to water sources, size of cemeteries, and number of burials. Additional factors such as coffin material, soil characteristics (e.g., lithology, mineralogy, grain size distribution, structure, thickness, leaching potential, permeability, plasticity, chemical properties, and presence of porous/fissured zones), topography, land use (e.g., presence of vegetation, agricultural practices, and urbanised areas), climatic characteristics (e.g., precipitation, temperature, and actual evapotranspiration), geological and hydrogeological aspects (e.g., groundwater flow mechanisms), abstraction rates, extension of source protection zones, depth of the water table, and seasonal fluctuations also play a crucial role in groundwater quality [2,11,14,17,31,35–38]. In cases where a cemetery is situated on permeable and porous soil, such as gravel or sand, leachate from decomposing corpses and coffin seepage can rapidly move and blend with the groundwater below [2]. Optimal decomposition requires homogeneous soils with balanced proportions of sand, silt, and clay (roughly 30% of each). During periods of high precipitation, intense runoff, and infiltration, when the water table is close to the soil surface, chemicals and pathogens can swiftly migrate to the groundwater [14]. It is advisable to assess cemeteries for potential risks using a comprehensive framework that takes into account risk significance, consequences, magnitude, and hazard identification [35].

Groundwater vulnerability is the tendency or likelihood of contaminants to reach a specific position in the groundwater system after their introduction at a location above the uppermost aquifer. The term first appeared in the 1970s [39] and gained notoriety in the 1980s [40]. It involves intrinsic vulnerability, which refers to the characteristics that affect the migration of pollutants towards groundwater [41], and specific vulnerability, which depicts the susceptibility to a specific contaminant or group of contaminants, considering aspects such as biogeochemical attenuation processes [42]. Because groundwater vulnerability cannot be measured directly [43], several indicators have been proposed to assess current groundwater quality or predict future scenarios [44,45]. Taghavi et al. [46] classified these evaluation methods into four categories: (i) overlay- and index-based methods [40,42,43], (ii) process-based simulation models [43,47], (iii) statistical methods (including orthodox and Bayesian methods) [42,43,48,49], and (iv) hybrid methods [50,51]. Other techniques have been put forth to assess the vulnerability of groundwater resources. These include the model of intrinsic groundwater vulnerability and specific vulnerability to pesticide

pollution [52,53], techniques for determining karst aquifer vulnerability [54], an approach that incorporates impact modelling, and an index-based approach to determine how vulnerable groundwater resources are to climate change [55].

The DRASTIC index proposed by Aller et al. [40] has already been used for groundwater vulnerability assessment in many studies [56–58], and it can be used to assess the risk of groundwater contamination associated with cemeteries. To reduce the subjectivity of the evaluation associated with the original model, modified or updated versions have been developed to identify appropriate ratings and determine weights for the DRASTIC parameters [58–61].

As cemeteries are sensitivity places with large spatial structures, they need to have a well-designed layout to allow funeral services [62–66]. Generating vulnerability DRASTIC maps involves handling substantial data, and GIS tools have been employed to manipulate hydrogeomorphological, hydrogeological, soil characteristics, and land-use data [67,68]. Map algebra calculations facilitate mathematical operations among thematic maps to generate composite spatial maps or charts, such as vulnerability or suitability maps [69]. GIS has been previously used to develop DRASTIC-based vulnerability maps, but mainly focused on contamination risks from wastewater facilities, garbage deposits, underground gas or fuel deposits, sanitary landfills, soils contaminated by industrial activities, and agricultural soils contaminated with an excess of fertilizers (specifically nitrate) or pesticides [45,51,57,70–72]. For example, Sinan and Razack [73] evaluated the vulnerability of Marrakech's Haouz aquifer to various pollution sources, including Marrakech's industrial park, industrial facilities, cemeteries, and waste deposits near Ourika and Tahanaout.

The main goal of this study was to develop a DRASTIC index-based vulnerability map for assessing the risk of aquifer contamination associated with two cemeteries in the Soure Region (Portugal) using GIS interpolation tools. Using the DRASTIC method [40], a geological analysis of the study area was conducted in two phases: the first phase considered the index independently of the pollutant load, and the second phase was developed based on the locations of particularities, resulting in a Final Classification that was considered the Specific DRASTIC [74]. The main innovation of this study is the use of this methodology to create maps that can be useful for defining measures to avoid groundwater pollution from cemeteries, both in existing spaces and new spaces. The DRASTIC approach was selected because it is the easiest to use and fits well with the GIS framework. Moreover, the approach is a computationally efficient model as it eliminates the need for intricate numerical analysis or multi-parameter simulation processes. What is more, though, is that it produces excellent results with little application cost. Due in part to the large number of input data points used, this methodology improves evaluation performance, thereby reducing the impact of errors on the final product.

*Design of Cemeteries in Portugal*

The construction of new public cemeteries was mandated by a decree dated 9 August 1834 [75]. As a result of the high number of deaths in the Portuguese Civil War (1832–1833) and during the cholera epidemic of 1833, the method of burying bodies in the ground was finally regulated [75]. The Decree of 21 September 1835 established that municipal authorities should allocate an area of land for the construction of cemeteries in all urban areas (villages, towns, and cities), but located at a safe distance to avoid contamination and health problems. With the new law, bodies had to be buried for 5 years in a pit made of individual soil at least 1.1 m deep and at least 0.33 m apart between graves. In 1962, Decree No. 44,220 appeared, defining the type of soil for burial: siliceous limestone, clayey limestone, or siliceous clayey, and the area must be designed for 50 years. The use of metal (zinc or lead) and solid wood coffins was prohibited to allow the bodies to degrade within five years, with the bones being able to be removed or buried deeper in the ground. Six years later (in 1968), the use of 20 L and 80 L of hydrated lime in wooden and metal coffins, respectively, was authorised to accelerate the decomposition of bodies. Graves and crypts that were unused or unmaintained for 10 years were transferred to the management

of local authorities. In 1982, a new law was published (Decree-Law No. 274/82) with instructions on how to bury or cremate mortal remains.

The 1998 law (Decree-Law no. 411/98) authorises that only zinc coffins can be buried in a crypt and prohibits the burial of bodies in mass graves, unless the law is revoked for special cases. The ashes of incinerated bodies can be kept in a burial urn, ossuary, or crypt or kept in the care of a family member. The graves must remain closed for at least five years, and the period may be extended if the remains are not degraded. Completely decomposed human remains can be transferred to an ossuary or a family grave or even be cremated at the request of relatives. However, if the bodies are not decomposed at the time of exhumation, they should remain closed until the skeletonization process is complete [75].

## 2. Materials and Methods

### 2.1. Location of Cemeteries and the Study Area

This work involved the study of two cemeteries (Samuel, with identification UC9; and Vinha da Rainha, with identification UC10), both in the municipality of Soure (central region of Portugal) (Figure 1). As no other important sources of anthropogenic pollution are known in the vicinity, these units represent the most serious threat to the quality of groundwater and public health from pollution. Whereas UC9 is situated higher up and farther away from the urban agglomeration, UC10 is situated almost flatly and nearer to urbanisations, where it may initially pose a greater risk of contamination. Despite their approximate linear distances of 2.5 Km (UC10) and 3.3 Km (UC9) from the hydromineral resource, Bicanho Spa, it may still be necessary to investigate the possibility of contamination.

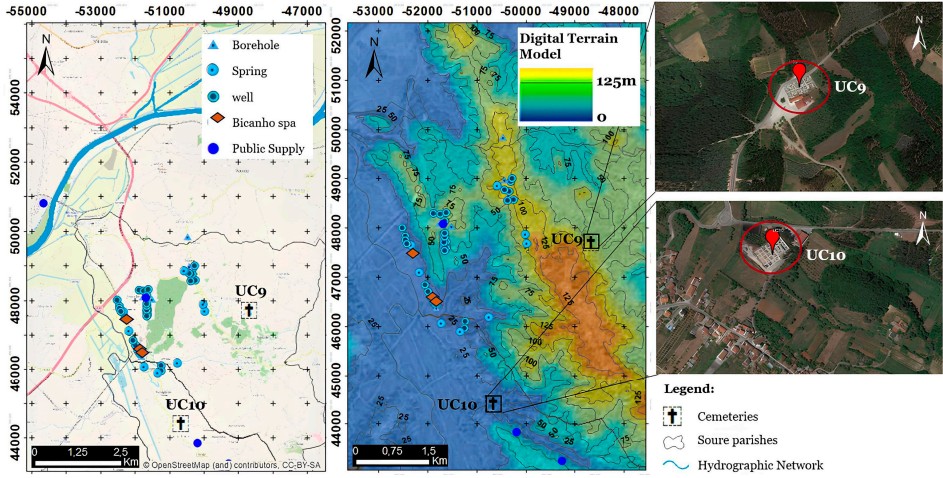

**Figure 1.** Geographic locations of two cemeteries in the Soure region.

### 2.2. Assessment of Groundwater Vulnerability

An assessment procedure consisting of three steps was devised to evaluate the possible effects of runoff from cemeteries to groundwater in the Soure area. The first step involved delineating the GWPZs. This research aims to evaluate the vulnerability of groundwater pollution using the DRASTIC index model and Specific DRASTIC technique, which was performed during the second and third phases, respectively.

#### 2.2.1. Mapping of GWPZs

In this section of the study, GWPZs were defined based on a variety of geological, hydrogeological, and environmental factors using RS, GIS, and multi-criteria decision analysis (MCDA) [76–79]. Pairwise comparisons can be used to solve complex decision-making problems by applying the AHP [79]. Figure 2 shows a flowchart that creates GWPZs using GIS. Ten thematic maps were reclassified (Figure 3): Geology, Slope, Lineament density, Drainage density (Dd), Precipitation, Land-Use/Land-Cover (LULC), Topographic Wetness

Index (TWI), Stream Power Index (SPI), Distance to rivers, and Normalised Difference Vegetation Index (NDVI).

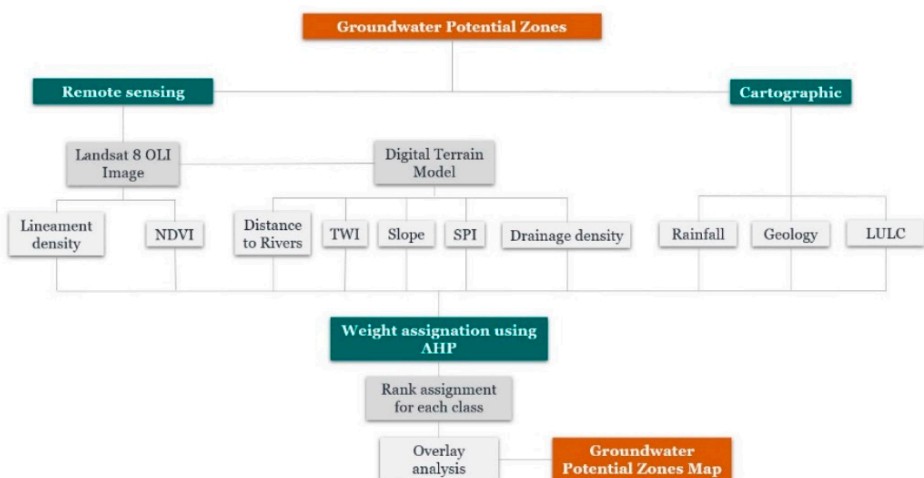

**Figure 2.** Flowchart that uses GIS to create a GWPZ map.

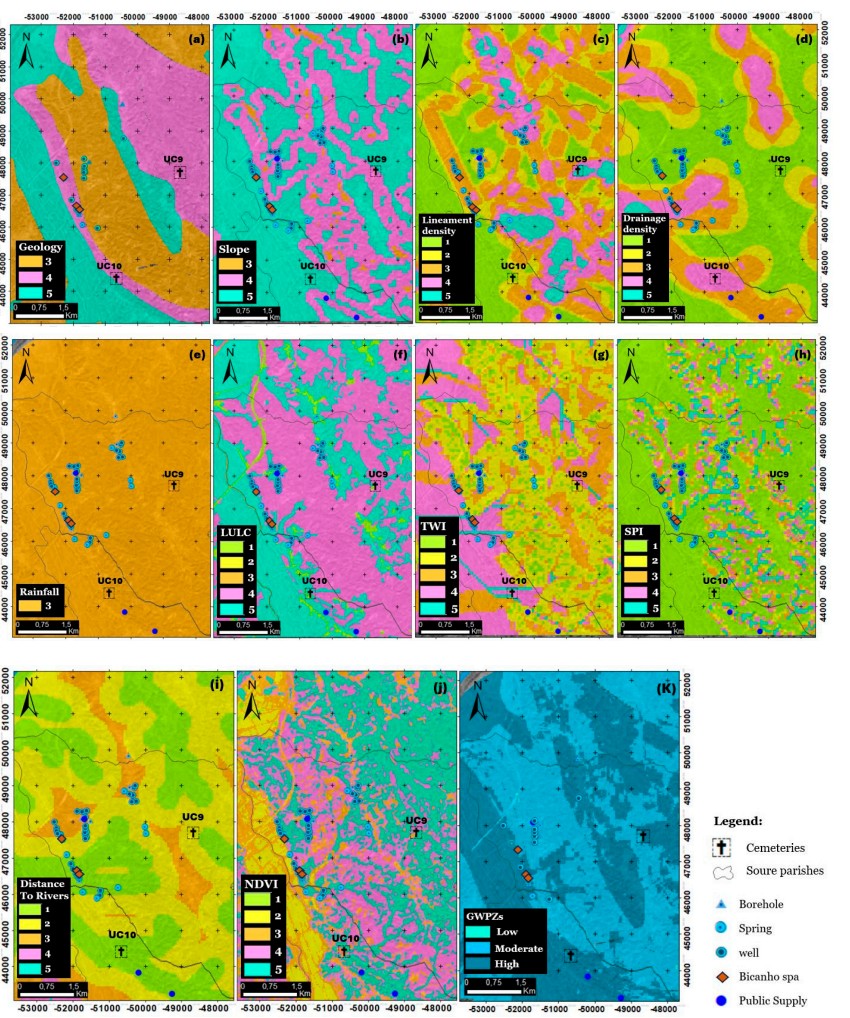

**Figure 3.** (**a**) Geological reclassification map; (**b**) Slope reclassification map; (**c**) Lineament density reclassification map; (**d**) Dd reclassification map; (**e**) Rainfall reclassification map; (**f**) LULC reclassification map; (**g**) TWI reclassification map; (**h**) SPI reclassification map; (**i**) Distance to rivers reclassification map; (**j**) NDVI reclassification map; (**k**) GWPZ map.

The delineation of the GWPZ map is a complex process because different environmental, climatic, and topographical factors are not widely understood [80,81]. The development of RS and GIS technologies has facilitated the delineation of large-scale GWPZs [82,83].

The different datasets used in the study to compute the GWPZs are detailed in Table 1. Although the spatial resolution of the satellite images provided by ESA (European Space Agency)-Sentinel-2 is generally higher, this implies more clarity and detail but also more data and storage. The Landsat 8 satellite is distinguished by the presence of thermal bands as well as band-8 (panchromatic), which is useful for improving image spectral resolution, and data are distinguished by a high radiometric resolution (16 bits), allowing the measurement of subtle variations in surface conditions.

**Table 1.** Data used for creating GWPZ input data.

| Data Type | Source | Format | Cell Size | Date | Used to Produce |
|---|---|---|---|---|---|
| DEM | USGS | Raster | | 2022 | Lineament density, NDVI, DTM—Distance to Rivers, TWI, Slope, SPI, Drainage density |
| Rainfall | SNIAMB | Shapefile polygon (1:1,000,000) converted to raster | 30 × 30 m | 1931–1960 | Annual precipitation—Recharge |
| Geology | LNEG | Shapefile polygon (1:500,000) converted to raster | | 1992 | Geology |
| LULC | DGT | Shapefile polygon (1:25,000) converted to raster | | 2018 | LULC |

Note: DEM—Digital Elevation Model; USGS—United States Geological Survey; DTM—Digital Terrain Model; SNIAMB—'Sistema Nacional de Informação de Ambiente'; LNEG—'Laboratório Nacional de Energia e Geologia'; LULC—Land use/Land cover; DGT—Direção Geral do Território.

Each raster was normalised using the geometric mean criteria following the evaluation of weights using the AHP method. For every feature class, a rating value between 1 and 5 (meaning "very low", "low", "medium", "high", and "very high") was assumed [84]. The rating values represented the suitability of the groundwater potential [84–89]. Table 2 displays each class's normalised weight and normalised rank for each variable.

The geological characteristics play a crucial role in determining groundwater potential because the hydraulic properties of the rock regulate the infiltration and percolation of water [90]. The geological map of the study region was converted from vector to raster format, and three categories were created once weight and rank had been assigned (Table 2, Figure 3a): (3) Taveiro sandstones and clays, Boa Viagem sandstones, and Carrascal sandstones; (4) Cabaços limestones and marls, Cabo Mondego limestones and marls and Costa de Arnes' crowded limestones; (5) Alluvium and sands and clays with kaolinite. Sedimentary rocks, such as limestones, possess substantial potential for storing groundwater.

Slopes in each area directly affect the rate of infiltration and also surface runoff, which in turn affects the recharge of groundwater, which is impacted by topography and/or slope gradient [91,92]. Steep slopes decrease infiltration and groundwater recharge because they allow less water to remain on the surface for longer periods due to rapid runoff. At the same time, because of their high rates of infiltration and low runoff, flat areas are better suited for recharge [93]. The slope map (degrees) was produced by using the Digital Elevation Model (DEM) and ArcMap's "Slope Tool". The study area's slope led to the creation of five categories: (1) >30, (2) 15–30, (3) 8–15, (4) 2–8, and (5) 0–2 (Table 2, Figure 3b).

**Table 2.** Values taken for normalised weights and thematic layer classifications.

| Variable | Units | NLW | % | Classes | Class Rank | NCR |
|---|---|---|---|---|---|---|
| Geology | - | 0.296 | 29.6 | Alluvium | 5 | 0.17 |
| | | | | Sands and clays with kaolinite | 5 | 0.14 |
| | | | | Taveiro sandstones and clays | 3 | 0.10 |
| | | | | Carrascal sandstones | 3 | 0.14 |
| | | | | Costa de Arnes' crowded limestones | 4 | 0.14 |
| | | | | Boa Viagem sandstones | 3 | 0.10 |
| | | | | Cabaços limestones and marls | 4 | 0.10 |
| | | | | Cabo Mondego limestones and marls | 4 | 0.10 |
| Slope | degree | 0.218 | 21.8 | 0–2 | 5 | 0.33 |
| | | | | 2–8 | 4 | 0.27 |
| | | | | 8–15 | 3 | 0.20 |
| | | | | 15–30 | 2 | 0.13 |
| | | | | >30 | 1 | 0.07 |
| Lineament density | Km/Km$^2$ | 0.131 | 13.1 | 0–0.49 | 1 | 0.07 |
| | | | | 0.49–1.34 | 2 | 0.13 |
| | | | | 1.34–2.18 | 3 | 0.20 |
| | | | | 2.18–3.23 | 4 | 0.27 |
| | | | | >3.23 | 5 | 0.33 |
| Drainage density (Dd) | Km/Km$^2$ | 0.108 | 10.8 | 0–0.31 | 1 | 0.07 |
| | | | | 0.31–0.88 | 2 | 0.13 |
| | | | | 0.88–1.53 | 3 | 0.20 |
| | | | | 1.53–2.40 | 4 | 0.27 |
| | | | | >2.40 | 5 | 0.33 |
| Rainfall | mm/year | 0.090 | 9.0 | 0–298 | 1 | 0.07 |
| | | | | 298–740 | 2 | 0.13 |
| | | | | 740–1100 | 3 | 0.20 |
| | | | | 1100–2070 | 4 | 0.27 |
| | | | | >2070 | 5 | 0.33 |
| Land-use/Land-cover (LULC) | - | 0.046 | 4.6 | Urban Area | 1 | 0.07 |
| | | | | Bare Ground | 2 | 0.13 |
| | | | | Water | 3 | 0.20 |
| | | | | Vegetation | 4 | 0.27 |
| | | | | Agricultural | 5 | 0.33 |
| Topographic Wetness Index (TWI) | (%) | 0.028 | 2.8 | 0–5.95 | 1 | 0.07 |
| | | | | 5.95–8.89 | 2 | 0.13 |
| | | | | 8.89–11.84 | 3 | 0.20 |
| | | | | 11.84–14.76 | 4 | 0.27 |
| | | | | >14.76 | 5 | 0.33 |
| Stream Power Index (SPI) | (%) | 0.028 | 2.8 | 0–5.68 | 1 | 0.07 |
| | | | | 5.68–11.36 | 2 | 0.13 |
| | | | | 11.38–21.33 | 3 | 0.20 |
| | | | | 21.33–57.11 | 4 | 0.27 |
| | | | | >57.11 | 5 | 0.33 |
| Distance to Rivers | (m) | 0.031 | 3.1 | 0–138.45 | 5 | 0.33 |
| | | | | 138.45–332.27 | 4 | 0.27 |
| | | | | 332.27–567.63 | 3 | 0.20 |
| | | | | 567.63–858.37 | 2 | 0.13 |
| | | | | <858.37 | 1 | 0.07 |
| NDVI | - | 0.024 | 2.4 | −1–0.02 | 1 | 0.07 |
| | | | | −0.02–0.09 | 2 | 0.13 |
| | | | | 0.09–0.22 | 3 | 0.20 |
| | | | | 0.22–0.31 | 4 | 0.27 |
| | | | | >0.31 | 5 | 0.33 |

Note: NLW—Normalised Layer Weight; NCR—Normalised Class Rank.

Lineaments, characterised by their straight or nearly straight form, are prominent land features that are accentuated by the permeability of the soil and are widespread across the Earth's surface [94,95]. Intrinsic permeability and porosity can be used to broadly characterize underlying fractures, faults, or joints [96,97]. The movement and storage of groundwater as well as the facilitation of water infiltration into the subsurface depend on the lineaments [98]. Following extraction, lineament discontinuities were examined using Landsat images on ArcGIS, and the "Line Density Tool" was used to create a lineament density map (Km/Km$^2$). Based on natural breaks, the following five categories were established: (1) 0–0.49, (2) 0.49–1.34, (3) 1.34–2.18, (4) 2.18–3.23, and (5) >3.23 (Table 2, Figure 3c).

Drainage is a mechanism that has an important role in controlling the hydrogeological characteristics of soils [85], and drainage density is defined as the surface area of a drained basin divided by the total length of its watercourses [99]. The groundwater recharge volume is correlated with the overall length of the drainage densities [84], and a zone with a high drainage density contributes significantly to surface runoff while retaining relatively little groundwater [100]. However, the drainage system is affected by several variables, including topography, climate, slope gradient, rainfall, vegetation cover, subsurface features [96], and the type and structure of the bedrock [101]. This variable makes it easier to understand and assess data about groundwater infiltration, permeability, runoff potential, and relief by providing a suitable numerical measurement [94]. The drainage density (Km/Km$^2$) was determined by using Equation (1) in conjunction with the Stream Network and the Line Density Tool [82].

$$D = \sum_{i-1}^{n} \frac{Di}{A} \tag{1}$$

where ($A$) represents the basin area (Km$^2$) and ($Di$) is the total length of all streams in stream order $i$ (Km). The "Hydrology Tool" in ArcMap, along with the Fill DEM, Flow Direction, Flow Accumulation, Stream Order, and Stream to Feature procedures, was used to create the Stream Network. Based on natural breaks, five categories were established: (1) 0–0.25, (2) 0.25–1.02, (3) 1.02–1.79, (4) 1.79–2.56, and (5) >2.56 (Table 2, Figure 3d).

Rainfall is a hydrologic process that restores aquifers, and it is a major factor in determining groundwater potential [86]. Although more recent total precipitation data were calculated at the study site, data from 1931 to 1960 [102] were used in the GIS environment because they were available in polygon shapefile format and the most recent data were contained within the polygon. Based on natural breaks, the study area's mean annual rainfall intensity was split into five zones: (1) 0–298, (2) 298–740, (3) 740–1100, (4) 1100–2070, and (5) >2070 (Table 2, Figure 3e).

LULC significantly influences how groundwater recharge occurs [103]. Plants and trees can store water in their leaves and stems and allow it to enter the earth through their roots and rhizomes, thus contributing to recharging groundwater. This circumstance leads to the demand for groundwater extraction on agricultural and plantation land. However, the increase in the use of concrete in urban areas leads to an increase in surface runoff and a decrease in recharge. The COS2018 chart [104] provided the LULC data, and five categories were created: (1) Urban Area, (2) Bare ground, (3) Water bodies, (4) Vegetation, and (5) Agricultural (Table 2, Figure 3f).

The TWI map was created by Beve and Kirkby [105] and is the most often used map in hydrological studies [102,106]. The TWI's upslope area can be used to measure subsurface lateral transmissivity or as a local slope indicator [107,108]. Soil moisture content is one of the hydrological parameters that is significantly impacted by TWI in each area [109]. Because the zoning and extent of saturated areas affect the occurrence of springs [107], the higher the TWI, the greater the groundwater potential. TWI calculations [110] provide an overview of how foothill, hillslope, and topographic roughness affect lateral groundwater flow. Equation (2) [111] was used to calculate TWI, which measures a cell's propensity

to retain water. It also makes it easier to find favourable locations with slow runoff and concentration.

$$\ln \frac{\alpha}{\tan b} \tag{2}$$

Based on natural breaks, the following five categories were established: (1) 0–5.95, (2) 5.95–8.89, (3) 8.89–11.84, (4) 11.84–14.76, and (5) >14.76 (Table 2, Figure 3g).

The SPI measures the effects of landform, elevation, and slope on groundwater resources and is a useful metric for identifying areas where groundwater infiltration occurs. The runoff influence increased with the SPI value, which was determined using slope and flow accumulation parameters in ArcGIS [90]. Quantile breaks were used to create the following five categories: (1) 0–5.68, (2) 5.68–11.36, (3) 11.36–21.33, (4) 21.33–57.11, and (5) >57.11 (Table 2, Figure 3h).

Because local alluvial layers are typically found near river courses and because sites along rivers are best-suited for effective infiltration and subsequent recharge of groundwater, the distance from hydrographic networks is significant in hydrogeological research [112]. Rivers contribute to groundwater potential zones within watersheds, which in turn affects them. To begin the distance categories, the Euclidean distance tool from the ArcGIS spatial analyst tools was utilised. Based on natural breaks, the following five categories were established: (1) >858.37, (2) 567.63–858.37, (3) 332.27–567.63, (4) 138.45–33.27, and (5) 0–138.45 (Table 2, Figure 3i).

The NDVI layer quantifies vegetation by measuring the difference between near-infrared light, which vegetation strongly reflects, and red light, which vegetation absorbs. It was created using ArcMap and Landsat 8 images, with water being the most likely result given the negative values. Conversely, there is a strong likelihood that it has dense green leaves if the NDVI value is near +1. On the other hand, a region with an NDVI close to zero is probably urbanised and lacks vegetation. Based on natural breaks, the following five categories were established: (1) −1–(−0.02), (2) −0.02–0.09, (3) 0.09–0.22, (4) 0.22–0.31, and (5) >0.31 (Table 2, Figure 3j).

The AHP approach was used to determine the weight of various layers. The first step was to create a Pairwise Comparison Matrix (PCM) (Table 3) using Saaty's (1–9) relative importance scale (Table 4) [113].

**Table 3.** Matrix for pairwise comparison of variables in the AHP method.

| Seven-Variable Pairwise Comparison Matrix for the AHP Method | | | | | | | | | | |
|---|---|---|---|---|---|---|---|---|---|---|
| Variable | Geology | Slope | Lineament Density | Dd | Rainfall | LULC | TWI | SPI | Distance to Rivers | NDVI |
| Geology | 1 | 2 | 3 | 4 | 7 | 8 | 8 | 8 | 7 | 7 |
| Slope | 0.500 | 1 | 3 | 2 | 5 | 5 | 8 | 8 | 7 | 7 |
| Lineament Density | 0.333 | 0.333 | 1 | 2 | 5 | 3 | 4 | 4 | 4 | 5 |
| Drainage Density | 0.250 | 0.500 | 0.500 | 1 | 3 | 3 | 4 | 4 | 4 | 5 |
| Rainfall | 0.143 | 0.200 | 0.200 | 0.333 | 1 | 4 | 5 | 5 | 3 | 7 |
| LULC | 0.125 | 0.200 | 0.333 | 0.333 | 0.250 | 1 | 3 | 3 | 2 | 1 |
| TWI | 0.125 | 0.125 | 0.250 | 0.250 | 0.200 | 0.333 | 1 | 1 | 1 | 2 |
| SPI | 0.125 | 0.125 | 0.250 | 0.250 | 0.200 | 0.333 | 1.000 | 1 | 1 | 2 |
| Distance to Rivers | 0.143 | 0.143 | 0.250 | 0.250 | 0.333 | 0.500 | 1.000 | 1.000 | 1 | 3 |
| NDVI | 0.143 | 0.143 | 0.200 | 0.200 | 0.143 | 1.000 | 0.500 | 0.500 | 0.500 | 1 |
| SUM | 2.887 | 4.769 | 8.983 | 10.617 | 22.126 | 26.167 | 35.500 | 35.500 | 30.500 | 40.000 |

**Table 4.** Saaty's scale of relative importance.

| Scale | Definition | Explanation |
|---|---|---|
| 1 | Equal significance | Each of the two activities contributes equally to the goal |
| 3 | moderate significance over the other | One activity is strongly preferred over another by experience and judgment |
| 5 | Essential or strong significance | One activity is favoured over another by experience and judgement |
| 7 | Very strong significance | An activity is highly preferred, and its practical dominance is evidenced |
| 9 | Extreme significance | The strongest possible order of affirmation is present in the evidence supporting one activity over another |
| 2, 4, 6, 8 | Values in the middle of the two close decisions | When a compromise is required |

A PCM for variables was produced by comparing each layer based on its relative importance (Table 2). To minimise the related subjectivity, the normalised weights were computed in the second step of this procedure. Equation (3) [114] was also used to calculate the sum of the values in each column, which is shown in Table 5.

$$L_{ij} = \sum_{n=1}^{n} C_{ij} \tag{3}$$

where $(C_{ij})$ represents the variable used in the analysis and $(L_{ij})$ represents the PCM's total column value.

**Table 5.** Normalised matrix for pairwise comparison of variables in the AHP method.

| Normalised Pairwise Comparison Matrix | | | | | | | | | | | | |
|---|---|---|---|---|---|---|---|---|---|---|---|---|
| **Variable** | **Geology** | **Slope** | **Lineament Density** | **Dd** | **Rainfall** | **LULC** | **TWI** | **SPI** | **Distance to Rivers** | **NDVI** | **Total** | **NWT** |
| Geology | 0.347 | 0.419 | 0.334 | 0.377 | 0.317 | 0.306 | 0.225 | 0.225 | 0.229 | 0.179 | 2.958 | 0.296 |
| Slope | 0.173 | 0.210 | 0.334 | 0.189 | 0.226 | 0.191 | 0.225 | 0.225 | 0.229 | 0.179 | 2.181 | 0.218 |
| Lineament Density | 0.115 | 0.070 | 0.111 | 0.189 | 0.226 | 0.115 | 0.113 | 0.113 | 0.131 | 0.128 | 1.311 | 0.131 |
| Drainage Density | 0.087 | 0.105 | 0.056 | 0.094 | 0.136 | 0.115 | 0.113 | 0.113 | 0.131 | 0.128 | 1.078 | 0.108 |
| Rainfall | 0.049 | 0.042 | 0.022 | 0.031 | 0.045 | 0.153 | 0.141 | 0.141 | 0.098 | 0.179 | 0.901 | 0.090 |
| LULC | 0.043 | 0.042 | 0.037 | 0.031 | 0.011 | 0.038 | 0.084 | 0.084 | 0.065 | 0.026 | 0.461 | 0.046 |
| TWI | 0.043 | 0.026 | 0.028 | 0.023 | 0.009 | 0.013 | 0.028 | 0.028 | 0.033 | 0.051 | 0.282 | 0.028 |
| SPI | 0.043 | 0.026 | 0.028 | 0.023 | 0.009 | 0.013 | 0.028 | 0.028 | 0.033 | 0.051 | 0.282 | 0.028 |
| Distance to Rivers | 0.049 | 0.030 | 0.028 | 0.023 | 0.015 | 0.019 | 0.028 | 0.028 | 0.033 | 0.051 | 0.304 | 0.031 |
| NDVI | 0.049 | 0.030 | 0.022 | 0.019 | 0.006 | 0.038 | 0.014 | 0.014 | 0.016 | 0.026 | 0.234 | 0.024 |

To create the Normalised Pairwise Comparison Matrix (NPCM), each column value was divided by the sum of the column values [87,114]. Each variable's normalised weight (NWt) was calculated by averaging all the values in the associated row of the NPCM (Table 5) [83,93]. Each normalised weight multiplied by all is equal to 1.

Because the AHP method is dependent on subjective or individual judgements, its application may lead to some inconsistencies [96]. The Consistency Ratio (*CR*) was computed to assess the accuracy. First, each PCM column was multiplied by the variable weight. The weighted sum value was then obtained by adding the values of each row. A division between the variable's weight and the weighted sum value was then performed, yielding a $\lambda$ value [87]. Equation (4) [115] can be used to determine the maximum eigenvalue ($\lambda$ *max*):

$$\lambda\ max = \frac{C1 + C2 + C3 \ldots Cn}{n} \tag{4}$$

The $(\lambda)$ values are $(C1)$ through $(Cn)$, and the number of criteria is $(n)$. A $(\lambda\ max)$ value of 11.283 was found in this study.

The value of the Consistency Index (*CI*) was then calculated using Equation (5) [115]:

$$CI = \frac{\lambda\ max - n}{n - 1} \tag{5}$$

where ($\lambda\ max$) is the judgement matrix's maximum eigenvalue and ($n$) is the number of criteria. This study yielded a *CI* value of 0.143.

Finally, Equation (6) was used to calculate the *CR* [115]:

$$CR = \frac{CI}{RI} \tag{6}$$

where, according to [115], ($RI$) denotes the Random Consistency Index and ($CI$) stands for *CI* (Table 6). A consistency ratio value of 1.51 was found in this investigation.

**Table 6.** Random Consistency Index (*RI*) values for n variables.

| N | 1 | 2 | 3 | 4 | 5 | 6 | 7 | 8 | 9 | 10 | 11 | 12 | 13 | 14 | 15 |
|---|---|---|---|---|---|---|---|---|---|----|----|----|----|----|----|
| *RI* | 0.00 | 0.00 | 0.58 | 0.90 | 1.12 | 1.24 | 1.32 | 1.41 | 1.45 | 1.51 | 1.52 | 1.54 | 1.56 | 1.58 | 1.59 |

If the CR is less than 0.10, the inconsistency is acceptable; if the CR is greater than 0.10, the judgements must be updated. The value 0.094 was determined to be a valid CR value in this investigation.

The GWPZ map was created by Equation (7), which integrates all parameters in order of significance using the Groundwater Potential Index (*GWPI*) [87].

$$GWPI = \sum_{W=1}^{m} \sum_{j=1}^{n} (Wj \times Xi) \tag{7}$$

($Wj$) is the normalised weight of the ($j^{\text{th}}$) variable, and ($Xi$) is the normalised weight of the variable's ($i^{\text{th}}$) class. The Raster Calculator Tool in ArcGIS was used to perform the corresponding integration.

Table 7 summarises the assigned normalised weights and ranks of thematic layers found for each cemetery.

**Table 7.** Assigned normalised weights of thematic layers.

| Cemetery | Geology | Slope | Line Density | Dd | Rainfall | LULC | TWI | SPI | Distance to Rivers | NDVI | GWPZ |
|----------|---------|-------|--------------|----|----------|------|-----|-----|--------------------|------|------|
| UC9 | 4 | 4 | 5 | 1 | 3 | 4 | 2 | 4 | 2 | 5 | Moderate |
| UC10 | 4 | 5 | 1 | 4 | 3 | 5 | 4 | 1 | 1 | 3 | Good |

The cartography created for the GWPZs will be used to map the areas where aquifer recharge is favoured, as well as to define the various indices in the R parameter used in the DRASTIC index.

### 2.2.2. Mapping of DRASTIC Index Vulnerability

To map the DRASTIC index vulnerability, seven thematic maps were created: depth to groundwater (D), net recharge (R), aquifer material typology (A), soil type (S), topography (T), impact of the vadose zone (I), and hydraulic conductivity (C) [40]. Each parameter was further separated into representative classes, each of which was assigned an index (i), as presented in Table 7, to correlate with the local hydrogeological characteristics (Equation (8)).

$$DI = Di \times Dw + Ri \times Rw + Ai \times Aw + Si \times Sw + Ti \times Tw + Ii \times Iw + Ci \times Cw \tag{8}$$

where $(D), (R), (A), (S), (T), (I), (C)$ are the hydrogeological parameters, (i) is the rating for the area being evaluated (1–10), and (w) is the weight of the factor (1–5) (Table 8) [40]. The weight (w) of each DRASTIC index parameter represents its relative importance to other attributes. The vulnerability of the aquifer to pollution increases with increasing DRASTIC index.

The procedures for creating the DRASTIC-based vulnerability map are presented in Figure 4. The adopted weights and indices were proposed by Aller et al. [40] and have already been successfully validated in other works [60,69,70,116–122].

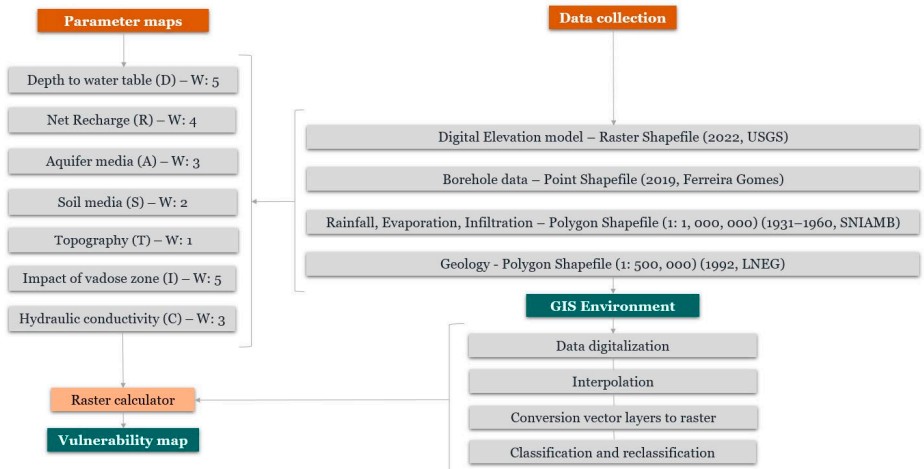

**Figure 4.** Flowchart that uses the GIS DRASTIC index to create a groundwater vulnerability map.

The seven hydrogeological layers were overlayed to produce the DRASTIC vulnerability index map using the ArcGIS raster calculator. Table 9 displays the quantitative and qualitative classifications of aquifer pollution susceptibility, which are categories modified from the values presented in Hamza et al. [57] and LNEC. In Portuguese studies, this division is the most prevalent.

Equation (10) presents the computational procedure used to generate the DRASTIC-based vulnerability map. It was adapted from [68] and involved arithmetic operations of maps according to Equation (9) and the values in Table 8 to overlap the seven thematic maps. Equation (10) was used to generate the value of every cell in the vulnerability map by performing an arithmetic operation. Values were stored in every cell of each thematic map. To create the vulnerability map, Equation (8) was added to the raster calculator function.

$$\left( M_{ij}^{k} \right)_{mn} \times W = \sum_{k=1}^{tm} \left( \begin{pmatrix} M_{11}^{k} & M_{12}^{k} & \cdots & M_{1n}^{k} \\ M_{21}^{k} & M_{22}^{k} & \cdots & M_{2n}^{k} \\ \vdots & \vdots & \cdots & \vdots \\ M_{m1}^{k} & M_{m2}^{k} & \cdots & M_{mn}^{k} \end{pmatrix} \times W^{k} \right) \tag{9}$$

where $\left( M_{ij}^{k} \right)$ is the vector of cell values from each thematic map in line (i) and row (j), (m) and (n) are the dimensions of the thematic grid map, (k) is the thematic map, (tm) is the number of thematic maps, and (W) is the vector of values associated with each cell.

**Table 8.** Partial indices (Ip) used for calculating the DRASTIC index (DI) according to the various parameters and classes.

**Parameter — Partial Indices (PIs) Function of the Various Parameters and Their Classes**

**D**

| Depth (m) | <1.50 | 1.50–4.60 | 4.60–9.10 | 9.10–15.20 | 15.20–22.90 | 22.90–30.50 | >30.50 |
|---|---|---|---|---|---|---|---|
| Ip | 10 | 9 | 7 | 5 | 3 | 2 | 1 |

**R**

| Recharge (mm/year) | <51 | 51–102 | 102–178 | 178–254 | >254 |
|---|---|---|---|---|---|
| Ip | 1 | 3 | 6 | 8 | 9 |

**A**

| Aquifer material | clayey schist, clay-stone | metamorphic/igneous rock | metamorphic/igneous-altered rock | glacial deposits | sandstone, limestone, and claystone, stratified | sandstone | limestone | sand and gravel | basalt | carsified limestone |
|---|---|---|---|---|---|---|---|---|---|---|
| Ip | 1–3 | 2–5 | 3–5 | 4–6 | 5–9 | 4–9 | 4–9 | 4–9 | 2–10 | 9–10 |
| Ip Typical | 2 | 3 | 4 | 5 | 6 | 6 | 6 | 8 | 9 | 10 |

**S**

| Soil Type | thin or absent | gravel | sand | peat | consistent clay and/or expansible | sandy | loam | silty | clayey | muddy | non-expan. Clay |
|---|---|---|---|---|---|---|---|---|---|---|---|
| Ip | 10 | 10 | 9 | 8 | 7 | 6 | 5 | 4 | 3 | 2 | 1 |

**T**

| Slope (%) | <2 | 2–6 | 6–12 | 12–18 | >18 |
|---|---|---|---|---|---|
| Ip | 10 | 9 | 5 | 3 | 1 |

**I**

| Unsaturated zone | confining layer | clay/silt | clayey schist, claystone | limestone | sandstone | sandstone, limestone, and claystone, stratified | sand and gravel with many fines | metamorphic/igneous rock | sand and gravel | basalt | carsified limestone |
|---|---|---|---|---|---|---|---|---|---|---|---|
| Ip | 1 | 2–6 | 2–5 | 2–7 | 4–8 | 4–8 | 4–8 | 2–8 | 6–9 | 2–10 | 8–10 |
| Ip Typical | 1 | 3 | 3 | 6 | 6 | 6 | 6 | 4 | 8 | 9 | 10 |

**C**

| K (m/day) | <4.1 | 4.1–12.2 | 12.2–28.5 | 28.5–40.7 | 40.7–81.5 | >81.5 |
|---|---|---|---|---|---|---|
| Ip | 1 | 2 | 4 | 6 | 8 | 10 |

**Table 9.** Classes of vulnerability defined for DRASTIC [123].

| DRASTIC Index | |
| --- | --- |
| **Quantitative Classes** | **Qualitative Vulnerability** |
| 23–79 | Insignificant |
| 80–99 | Extremely low |
| 100–119 | Very low |
| 120–139 | Low |
| 140–159 | Average |
| 160–179 | High |
| 180–199 | Very high |
| 200–226 | Extremely high |

$$\left(S_{ij}\right)_{mn} = \begin{pmatrix} S_{11} & S_{12} & \cdots & S_{1n} \\ S_{21} & S_{22} & \cdots & S_{2n} \\ \vdots & \vdots & \cdots & \vdots \\ S_{m1} & S_{m2} & \cdots & S_{mn} \end{pmatrix} \tag{10}$$

where $\left(S_{ij}\right)$ is the vector of cell values for the suitability map in lines $(i)$ and $(j)$ and $(m)$ and $(n)$ are the dimensions of the suitability grid map.

2.2.3. Mapping of Specific DRASTIC Vulnerability

Changes were then introduced to the specific DRASTIC [124], considering the current groundwater abstractions in the territory, in two phases: (i) first, the lithological units were classified according to the classic DRASTIC index (DI), with the same values that define potential vulnerability, degree of vulnerability, and qualitative vulnerability class (Table 10); (ii) second, the various units were reclassified according to the location of water catchments and springs; factors considered: (a) presence of geological singularities (OGSs), such as lithological contacts, veins, faults, and fractures, with real or potential connection to water catchments and aquifers; (b) location of the geological unit (LGU) in relation to water catchments and springs. Depending on the circumstances listed below, detailed reclassification may result in a higher or lower degree of classification; the final classification was identified as Specific DRASTIC to differentiate it from the typical situation. Several scenarios were considered for the OGS Factor: (i) the unit maintains the vulnerability class under the general DRASTIC index (DI) if there were no discontinuities with an actual or potential connection to the water abstractions and aquifers; (ii) if there were discontinuities or springs connected to the aquifer, these locations were classified as "very high" to "extremely high" vulnerability (G = 7 to 8); (iii) if there were discontinuities or locations with the potential for springs, these areas were classified as "medium" to "high" vulnerability (G = 5 to 6, Table 10). The following scenarios were considered when it came to the LGU Factor: (i) if the unit was located upstream of areas that either currently or potentially discharge water (groundwater abstractions and springs), it must maintain its vulnerability class under the general DRASTIC index (DI); (ii) if the unit was located downstream of established or prospective areas of natural groundwater discharge, the unit's class must be lower than the overall DRASTIC index (DI). In certain cases, the unit class may even go from high vulnerability to low vulnerability, depending on the details of each case. Units with a very high vulnerability index may not be accessible to other types of occupation and, as long as the quality of underground resources is maintained, it is often necessary to use the entire area.

**Table 10.** Drastic index and associated vulnerabilities [124].

| Normal DRASTIC Index [40] | Potential Vulnerability (%) | Degree | Qualitative Vulnerability |
|---|---|---|---|
| <80 | <30 | 1 | Nonexistent |
| 80–99 | 30–39 | 2 | Very very low |
| 100–119 | 40–49 | 3 | Very low |
| 120–139 | 50–59 | 4 | Low |
| 140–159 | 60–69 | 5 | Moderate |
| 160–179 | 70–79 | 6 | High |
| 180–199 | 80–89 | 7 | Very high |
| >199 | >90 | 8 | Extremely high |

When applying the methodology to the case study, the main objective was to maintain good water quality at the different extraction points around the cemeteries.

## 3. Results and Discussion

### 3.1. Development of Maps Depicting Site Characteristics

Parameter D affects the extent and degree of physical and chemical attenuation and degradation, as well as the degree of interaction between subsurface constituents and percolating pollutants. Parameter D was estimated using lithology and correlation with studies carried out in that area [124,125].

Parameter R signifies the volume of water that infiltrates through the ground surface and reaches the water table within a specified land area The expected recharge rate was computed by the Thornthwaite method [126] and, additionally, it will be connected to the GWPZ map.

Geotechnical properties are intricately connected to parameter A, which denotes the attenuation potential based on the lithology within the saturated zone. The geological map of Portugal, sourced from LNEG at a scale of 1:500,000 [127], supplied the necessary information for computing the partial indices A, I, and C. The importance of parameter I in determining vulnerability is due to its impact on the residence time of pollutants in the unsaturated zone, and consequently, the probability of attenuation. The capacity of the aquifer to transport water, as indicated by parameter C, affects both the hydraulic gradient and the groundwater flow. High conductivity readings indicate a high risk of contamination. Singhal and Gupta's abacus [128] was used to calculate this parameter.

A geological map of the study area is shown in Figure 5. With a geological history spanning approximately 180 million years, the Figueira da Foz region is distinguished by both the more recent geodynamic setting of the Cenozoic deposits and the numerous Mesozoic evolutionary stages of the Lusitanian Basin [129]. Stratigraphic units within the study region are arranged in a substantial column extending from the Mesozoic (Upper Triassic) to the present, positioned discordantly over Precambrian and Paleozoic metasediments [129]. Cemetery UC9 is located within the Cabaços limestone and marl units, categorised within the middle to upper Oxfordian period. The brown or black fine-grained flint nodules are connected to the micro-sparitic and clayey micritic limestone decimetre scales deposited in freshwater limnic environments [130]. This unit has a thickness of about 250 m, attitudes of N20° W, 10–15° W; the limestone expresses itself more towards the base; this unit has poor aquifer suitability despite being quite fractured. Cemetery UC10 is in the Costa de Arnes crowded limestone unit, which consists of marly limestones, limestone sandstones, and marls with a lapped surface that is concreted or piled [130]. This unit has a thickness of 50 to 60 m, with a semi-parallel attitude to the previous unit; the base is composed of marls with detrital components, acquiring aquifer–aquitard characteristics. Clayish soils with a high specific surface area and cation-exchange capacity (CEC) are the most common because they maximise the retention of fluids and metals [131]. For a few reasons, limestone aquifers are especially susceptible to pollution namely due to the karst morphology that

they are prone to develop. Sinkholes and sinking streams are excellent ways for pollutants to seep from the topsoil into an underlying aquifer.

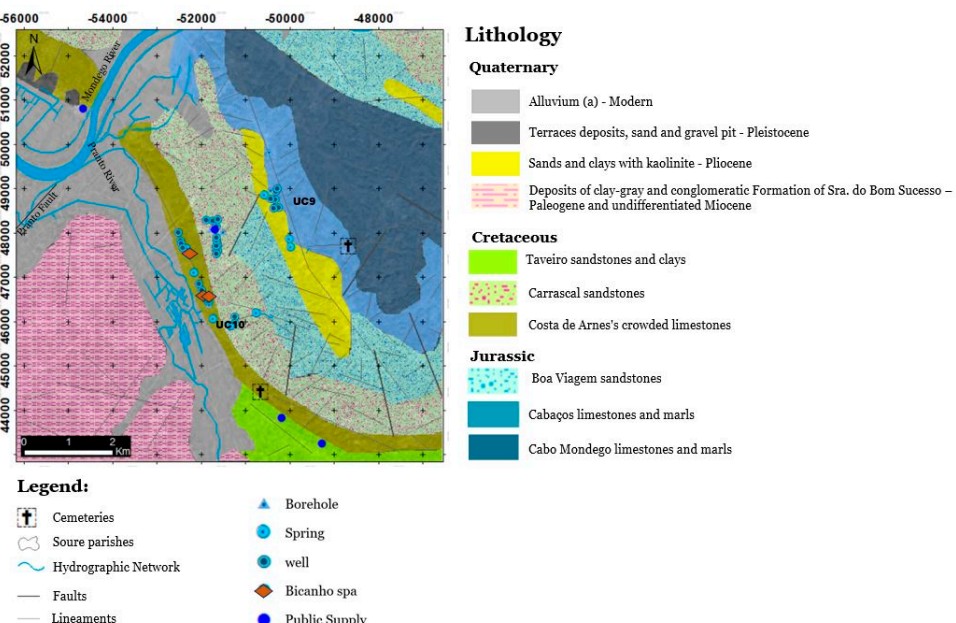

**Figure 5.** Geological map of the study area (adapted from [132]).

The cemeteries are in the Mondego River Basin. One of the Mondego River's tributaries on the left bank, the Pranto River, forms the western boundary of the municipality of Soure. It is distinguished by a low-lying area with elevations below 50 m until it joins the Mondego River at a height of roughly 2 m, known as the Vale do Pranto [124,132]. The river basin receives water from multiple streams and flows over alluvium that has been deposited on clays, marls, and limestones [124,129]. The extent of the fluctuations in water levels between the wet and dry seasons, along with the significant alterations in the flow of the springs through which they discharge, suggests that the self-regulating capacity of the karst aquifer systems is limited. There could be a 50–60% infiltration rate [132]. UC10 and the Bicanho Medical Spa are situated in the same lithological unit. Cemetery UC9 is in the 'Orla Ocidental Indiferenciada da Bacia do Mondego' (OOIBM) aquifer system (Figure 6). Cemetery UC10 is in the 'Figueira da Foz-Gesteira' aquifer system (Meso–Cenozoic) (Figure 7).

UC10 and the Bicanho Medical Spa are both located in the same aquifer system, as are UC9, numerous water wells, and some springs. The conceptual flow model of the 'Figueira da Foz-Gesteira' aquifer system (a) is essentially a geological volume composed primarily of porous detrital sediments that exhibit a diverse array of textures and lenticular structures. The system appears multi-layered because the clayey layers divide the multiple aquifer units. Owing to the wide range of granulometric compositions, hydraulic properties can vary significantly between sites. Karstification also affects the transmissive and storage capacities. These aquifers are especially susceptible to pollution owing to infiltration and rapid flow through karst structures. They also have a very low capacity for self-cleaning and the rapid spread of pathogens. A conceptual model (Figure 7) proposed by Portugal Ferreira [125] for the study area suggests that recharge occurs to the NE and at higher elevations, particularly in the Cabo Mondego limestone and marl units, and then evolves to the SW until the Pranto Fault.

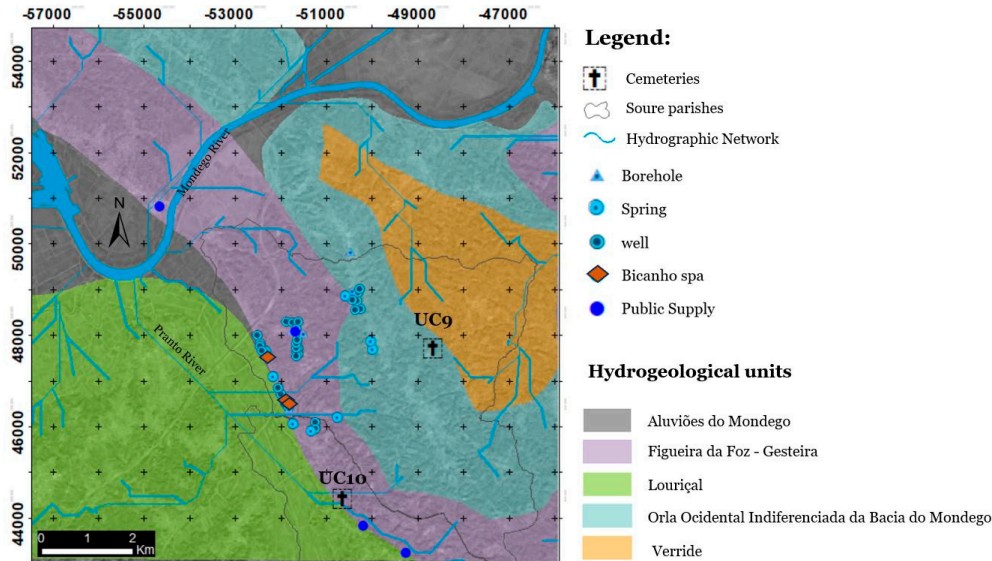

**Figure 6.** Hydrogeological map of the study area.

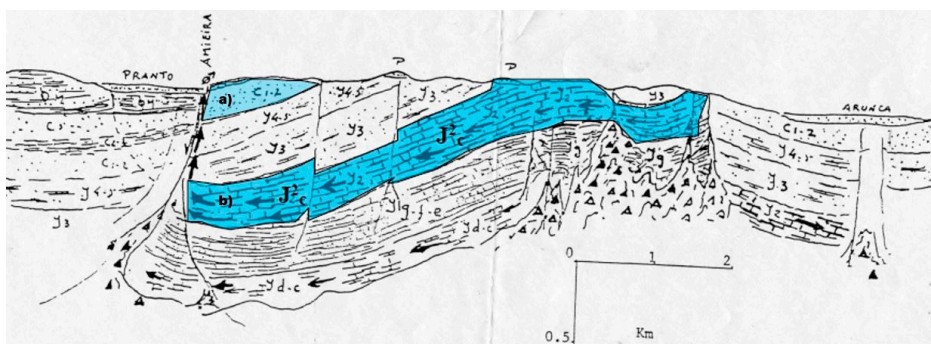

**Figure 7.** Structures representing the aquifer systems in the Bicanho-Amieira region: (**a**) Figueira da Foz-Gesteira aquifer system $C_{1-2}$; (**b**) Verride aquifer system—$J^2_c$ [124,125].

The cemeteries are located near the Atlantic coast and nested within the climate region of the west coast. Their Köppen Geiger classification is Csb, meaning that their climate is mesothermal (humid temperate) with a long and hot dry season in July. This climate is typical of the Mediterranean region owing to the influence of the ocean [133]. The coastal climate of the Mondego Basin is classified as type C2 B'2 according to the Thornthwaite climate classification [126] and it becomes wetter as the height of the basin increases. Using information gathered from the Portuguese Climate website [133], the Thornthwaite method (Figure 8) was used to determine the region's actual annual evapotranspiration. The air temperature in the study area ranges from 14.2 °C to 29.1 °C, with an average annual precipitation of 852.4 mm and actual evapotranspiration of 587.9 mm. The hydrological balance results led to the following conclusions: there is a dry period and a wet period. The first, known as the wet period, is represented by the water deficit (DH), which lasts from May/June to September, and the second, known as the water surplus (SH), extends from November to May. The water surplus is divided into two components: surface runoff (R) and underground runoff (G), resulting in SH = R + G = 264.5 L/m$^2$, which is initially very modest due to the contribution from underground recharge.

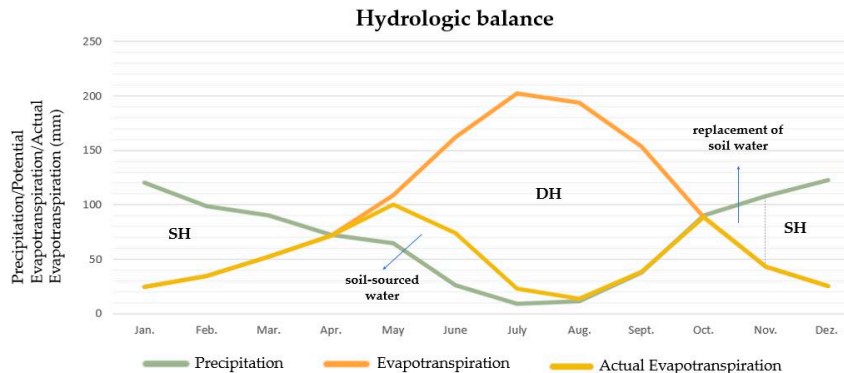

**Figure 8.** Monthly hydrological balance in the study area.

Parameter S assesses soil characteristics in the upper weathered zone to minimize the risk of pollution. Despite the endogenous features of cadavers influencing decomposition (e.g., age at death, cause of death, and fat content), concerns about burial soil and its impact on human taphonomy remain significant. The soil comprises an aqueous phase with dissolved elements, a gas phase, as well as biological and solid components, encompassing both organic and inorganic materials. Ferreira Gomes [124] looked closely at the data regarding each unit's soil type. The soil of both cemeteries is clay loam. This soil has a high potential for surface runoff when fully saturated. Permeability, or the ability of water to pass through soil, is either low or very low. Usually comprising less than 50% sand and more than 40% clay, they have a clayey texture. In some areas, they might also have a high potential for contraction and expansion. All soils that are less than 50 cm deep to a restrictive layer and all soils that have a groundwater table within the first 60 cm of depth were included in this group [132]. Soils with a range of intermediate characteristics, such as clayey sand and sandy clay [134–136], are best-suited for cemetery locations.

Changes in slope affect the T parameter as they influence drainage patterns. Flatter regions have become vulnerable to contaminant flows that can reach aquifers. Using topographic data from the USGS [137] and a previously prepared DTM, the slope map (Figure 9) was developed, which delineates zones suitable for aquifer recharge and prone to infiltration of pollutants. According to the slope map, cemetery UC9 has a higher slope (6–12%) than cemetery UC10 (<2%).

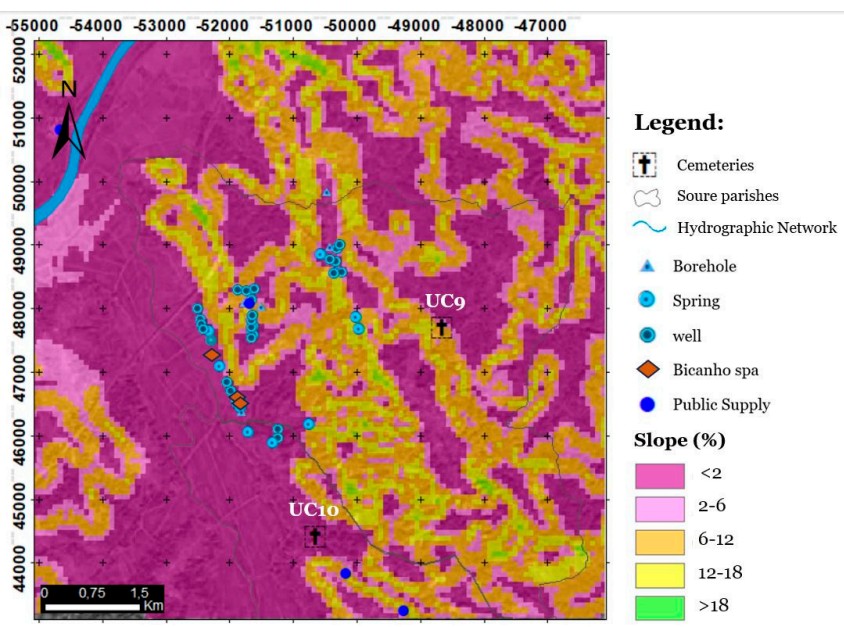

**Figure 9.** Slope map (%) of the study area.

### 3.2. Development of the Thematic Maps and the DRASTIC-Based Vulnerability Map

Based on the data presented in Table 11, seven thematic maps (Figure 10) were created and reclassified for each DRASTIC parameter.

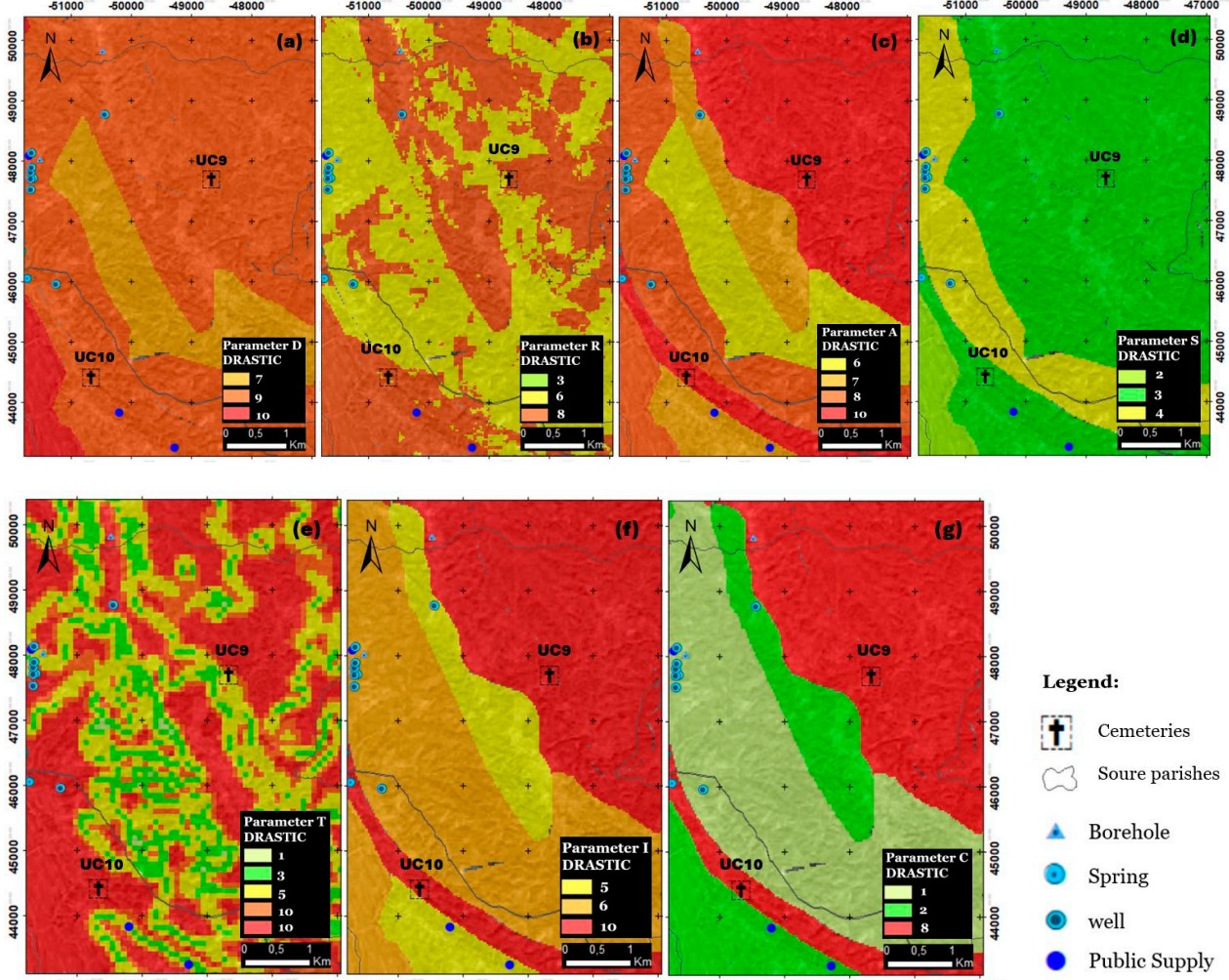

**Figure 10.** (**a**) Parameter D map; (**b**) Parameter R map; (**c**) Parameter A map; (**d**) Parameter S map; (**e**) Parameter T map; (**f**) Parameter I map; (**g**) Parameter C map of the studied area for the DRASTIC model.

Parameter D assumes a rating of nine for both cemeteries because the unsaturated zone depth is between 1.5 and 4.6 m. The index is relatively high because possible pollutants can enter the aquifer due to the water table being relatively close to the surface (Table 11 and Figure 10a).

The hydrological balance computation and the GWPZ chart indicate that both cemeteries have an index of eight for parameter R. The two cemeteries were evaluated with a recharge rate of 178–254 mm/year because they are situated in lithological units of karst limestones, which are favoured for aquifer recharge (Table 11 and Figure 10b).

**Table 11.** Characteristics of lithological units for the development of thematic DRASTIC maps.

| Unit | Parameter | Class | Index | Weight | Partial Index | DRASTIC | Vulnerability |
|---|---|---|---|---|---|---|---|
| I—Recent alluvium (free aquifer) | D | <1.5 m | 10 | 5 | 50 | 148 | Pollution is usually moderate but can occasionally be very high; it spreads quickly in flooded areas and along gravel lenticles. |
| | R | 102–178 mm/year | 6 | 4 | 24 | | |
| | A | Sand and gravel with many fines | 8 | 3 | 24 | | |
| | S | Muddy | 2 | 2 | 4 | | |
| | T | <2% | 10 | 1 | 10 | | |
| | I | Sand and gravel with many fines | 6 | 5 | 30 | | |
| | C | 4.1–12.2 m/day | 2 | 3 | 6 | | |
| II—Taveiro sands and clays (Upper Cretaceous, free aquifer) | D | 1.5–4.6 m | 9 | 5 | 45 | 136 | In general, low, because clay minerals allow heavy-metal adsorption. |
| | R | 102–178 mm/year | 6 | 4 | 24 | | |
| | A | Sand and gravel with clay | 7 | 3 | 21 | | |
| | S | Clay loam | 3 | 2 | 6 | | |
| | T | 2–6% | 9 | 1 | 9 | | |
| | I | Sand and gravel with clay | 5 | 5 | 25 | | |
| | C | 4.1–12.2 m/day | 2 | 3 | 6 | | |
| III—Costa de Arnes crowded limestones (Upper Cretaceous, free aquifer) | D | 1.5–4.6 m | 9 | 5 | 45 | 197 | The presence of karst limestones makes the lithological unit that contains UC10 very vulnerable. |
| | R | 178–254 mm/year | 8 | 4 | 32 | | |
| | A | Karsified limestone | 10 | 3 | 30 | | |
| | S | Clay loam | 3 | 2 | 6 | | |
| | T | <2% | 10 | 1 | 10 | | |
| | I | Karsified limestone | 10 | 5 | 50 | | |
| | C | 40.7–81.5 m/day | 8 | 3 | 24 | | |
| IV—Carrascal Sandstones (Middle Cretaceous, free to confined/semi-confined aquifer) | D | 1.5–4.6 m | 9 | 5 | 45 | 159 | In general, average |
| | R | 178–254 mm/year | 8 | 4 | 32 | | |
| | A | Sand and gravel | 8 | 3 | 24 | | |
| | S | Silty loam | 4 | 2 | 16 | | |
| | T | 2–6% | 9 | 1 | 9 | | |
| | I | Sand and gravel with many fines | 6 | 5 | 30 | | |
| | C | <4.1 m/day | 1 | 3 | 3 | | |
| V—Sands and clays with kaolinite (Pliocene, free aquifer) | D | 1.5–4.6 m | 9 | 5 | 45 | 136 | In general, low, because clay minerals allow heavy-metal adsorption. |
| | R | 102–178 mm/year | 6 | 4 | 24 | | |
| | A | Sand and gravel with kaolinite | 7 | 3 | 21 | | |
| | S | Clay loam | 3 | 2 | 6 | | |
| | T | 2–6% | 9 | 1 | 9 | | |
| | I | Sand and gravel with kaolinite | 5 | 5 | 25 | | |
| | C | 4.1–12.2 m/day | 2 | 3 | 6 | | |
| VI—Cabaços Limestones and Marls (Upper Jurassic, free to confined/semi-confined aquifer) | D | 1.5–4.6 m | 9 | 5 | 45 | 192 | The presence of karst limestones makes the lithological unit that contains UC9 very vulnerable. |
| | R | 178–254 mm/year | 8 | 4 | 32 | | |
| | A | Karsified limestone | 10 | 3 | 30 | | |
| | S | Clay loam | 3 | 2 | 6 | | |
| | T | 6–12% | 5 | 1 | 5 | | |
| | I | Karsified limestone | 10 | 5 | 50 | | |
| | C | 40.7–81.5 m/day | 8 | 3 | 24 | | |
| VII—Cabo Mondego Limestones and Marls (Middle Jurassic, free to confined/semi-confined aquifer) | D | 1.5–4.6 m | 9 | 5 | 45 | 189 | The presence of karst limestones makes the lithological unit very vulnerable. |
| | R | 102–178 mm/year | 6 | 4 | 24 | | |
| | A | Karsified limestone | 10 | 3 | 30 | | |
| | S | Clay loam | 3 | 2 | 6 | | |
| | T | <2% | 10 | 1 | 10 | | |
| | I | Karsified limestone | 10 | 5 | 50 | | |
| | C | 40.7–81.5 m/day | 8 | 3 | 24 | | |
| VIII—Boa Viagem Sandstones (Upper Jurassic, free to confined/semi-confined aquifer) | D | 4.6–9.1 | 7 | 5 | 35 | 131 | In general, low |
| | R | 102–178 mm/year | 6 | 4 | 24 | | |
| | A | Sandstone, limestone, and claystone, stratified | 6 | 3 | 18 | | |
| | S | Clay loam | 3 | 2 | 12 | | |
| | T | 2–6% | 9 | 1 | 9 | | |
| | I | Sandstone, limestone, and claystone, stratified | 6 | 5 | 30 | | |
| | C | <4.1 m/day | 1 | 3 | 3 | | |

Pollutant dilution and dispersion were significantly impacted by saturation zone water content and net recharge. Parameter A is determined by the lithological material in the saturated zone; lithological unit III—Costa de Arnes crowded limestones (Upper Cretaceous, free aquifer), and lithological unit VI—Cabaços limestones and Marls (Upper Jurassic, free to confined/semi-confined aquifer), were assigned an index of 10, the maximum because they were classified as Karstified limestones, units that are extremely permeable and allow for much faster water flow within the saturated zone (Table 11 and Figure 10c). As a pollutant reservoir and filter, the soil solid phase's capacity to retain and release hazardous chemical species and microbes is essential. Both cemeteries received an index of three due to their location on clay loam soil. Clay, which is composed of smaller particles, has a greater surface area and can retain more water (Table 10 and Figure 10d).

The UC9 cemetery is situated in a convex area with medium slopes (6–12%), earning a rating of five on the T parameter. In contrast, the UC10 cemetery is situated in a flat area with a 2% slope, earning a rating of 10. Contaminants in UC10 can linger long enough on the surface to penetrate (Table 11 and Figure 10e).

In terms of parameter I, both the UC9 and UC10 cemeteries are in lithological units with a lithological index of 10, the highest level, because they are Karstified limestones with a very short contact time with the pollutant (Table 11 and Figure 10f).

Lastly, an index of eight was given to the UC9 and UC10 cemeteries concerning parameter C. For the respective units where the cemeteries are located, a hydraulic conductivity of approximately 40.7 to 81.5 m/day was estimated (Table 11 and Figure 10g).

The vulnerability map (Figure 11) was then produced using the raster calculator function and Equation (8), weights from Figure 4, ratings from Table 8, and GIS matrix operations through Equations (9) and (10). The study area's values ranged from 105 (extremely low vulnerability) to 197 (very high vulnerability). The cemeteries at UC9 and UC19 are in areas with values of 192 and 197, respectively, indicating extremely high vulnerability (Figure 11a). It is required that an environmental monitoring programme is implemented at cemetery UC10, akin to that described in Directive 1999/31/EC [138], for groundwater uses (e.g., wells, holes, springs, and hot springs). With less clay present, weaker, less purifying soils at the surface, and a location in a flat area where pollutants are more likely to seep into the aquifer, UC10 is more vulnerable than UC9. The following factors contributed to vulnerability: I > D > R > A > C > S > T.

The occupation rate map (Figure 11b) was constructed from the cemetery surface area (TSC), grave surface area (SAB) (typically 2.6 m × 1.5 m, length × width), and occupancy rate (SAB/TSC ratio) observed for 2014 in [139]. A flow direction map was also created to safeguard water quality (Figure 11c). Pedrosa et al. [139] note that UC9 (14.0%) has a marginally lower occupancy rate than UC10 (15.4%) for each cemetery (Figure 11b).

According to the flow direction tool, surface waters in UC9 flow to the north and in UC10 to the west (Figure 11c, blue arrows)Although the 500 m buffer applied to all georeferenced water points is still quite far from the cemeteries in question, it is important to remember that the shared aquifer units are quite close (Figure 11c). However, it is established that in this case, there is no reason for the cemeteries under study to be concerned about any water hole becoming contaminated. It is critical to remember that many homes have water extraction points that are not listed in any database.

Every lithological unit's unique DRASTIC analysis is displayed in degrees in Table 12, and the representation is shown in Figure 12. Because Unit I have no OGS and does not affect the locations of cemeteries or georeferenced water sources, it will move from degree 5 to 4. Unit II behaves similarly to Unit I, but because UC10 has a superficial flow to the west that is draining in that direction, it will keep the degree at 4, not drop it to 3. Considering that Unit III is a part of the same aquifer system as the mineral water resource at Bicanho Medical Spa, there are numerous particularities to consider. The OGS reports that there are no discontinuities or faults near the UC10. Meanwhile, the LGU notes that although the cemetery is situated downstream of the georeferenced water points, the grade of 7 will be preserved because of the significance of these same resources. Georeferenced water

points are present in Unit IV, but they are all upstream of Unit 10 itself. Even though there are a lot of discontinuities in the unit in question, none of them seem to be dangerous for moving potential contaminants from UC10. The grade of five will be upheld for security concerns. There are large discontinuities and some faults cross-unit V, but they do not affect where the cemeteries are located. Upstream, there is only one georeferenced water point, so the grade will drop from 4 to 3. Unit VI contains UC9 as well as a water point that provides a public supply upstream. The degree will persist because UC9 is in an area with northerly surface runoff. Because of how far away UC9 is from the water point, it has not been raised to a higher level. Unit VII features a georeferenced water point, discontinuity, and a few faults; however, because it is situated upstream of the cemeteries, the grade will drop from 7 to 6. Unit VIII is finally distinguished from the other units by a density of lineament and two clearly defined faults crossing it. Nevertheless, they do not affect the cemeteries' susceptibility to pollution. If the grade was dropped from 4 to 3, georeferenced water points were also protected.

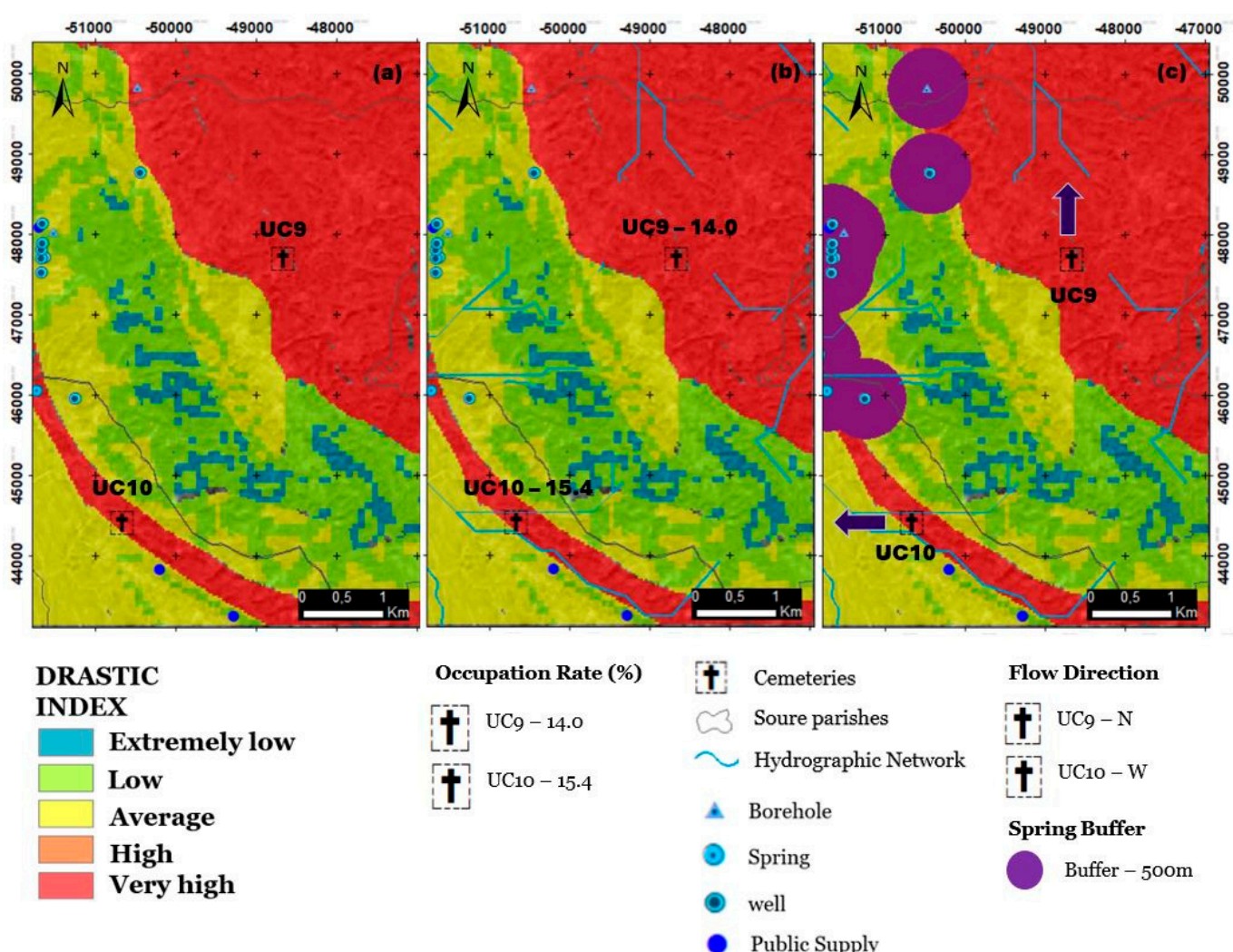

**Figure 11.** (**a**) Vulnerability map to pollution—Drastic index; (**b**) Occupation rate map; (**c**) Flow direction and buffer zones to water abstraction points of the studied area.

**Table 12.** Drastic index and Specific Drastic index in degrees.

| Unit | DRASTIC Index | Potential Vulnerability (%) | Degree | Qualitative Vulnerability | Specific Vulnerability Degree | Qualitative Vulnerability |
|---|---|---|---|---|---|---|
| I—Recent alluvium (free aquifer) | 148 | 60–69 | 5 | Moderate | 4 | Low |
| II—Taveiro sands and clays (Upper Cretaceous, free aquifer) | 136 | 50–59 | 4 | Low | 4 | Low |
| III—Costa de Arnes crowded limestones (Upper Cretaceous, free aquifer) | 197 | 80–89 | 7 | Very high | 7 | Very high |
| IV—Carrascal Sandstones (Middle Cretaceous, free to confined/semi-confined aquifer) | 159 | 60–69 | 5 | Moderate | 5 | Moderate |
| V—Sands and clays with kaolinite (Pliocene, free aquifer) | 136 | 50–59 | 4 | Low | 3 | Very Low |
| VI—Cabaços Limestones and Marls (Upper Jurassic, free to confined/semi-confined aquifer) | 192 | 80–89 | 7 | Very high | 7 | Very high |
| VII—Cabo Mondego Limestones and Marls (Middle Jurassic, free to confined/semi-confined aquifer) | 189 | 80–89 | 7 | Very high | 6 | High |
| VIII—Boa Viagem Sandstones (Upper Jurassic, free to confined/semi-confined aquifer) | 131 | 50–59 | 4 | Low | 3 | Very Low |

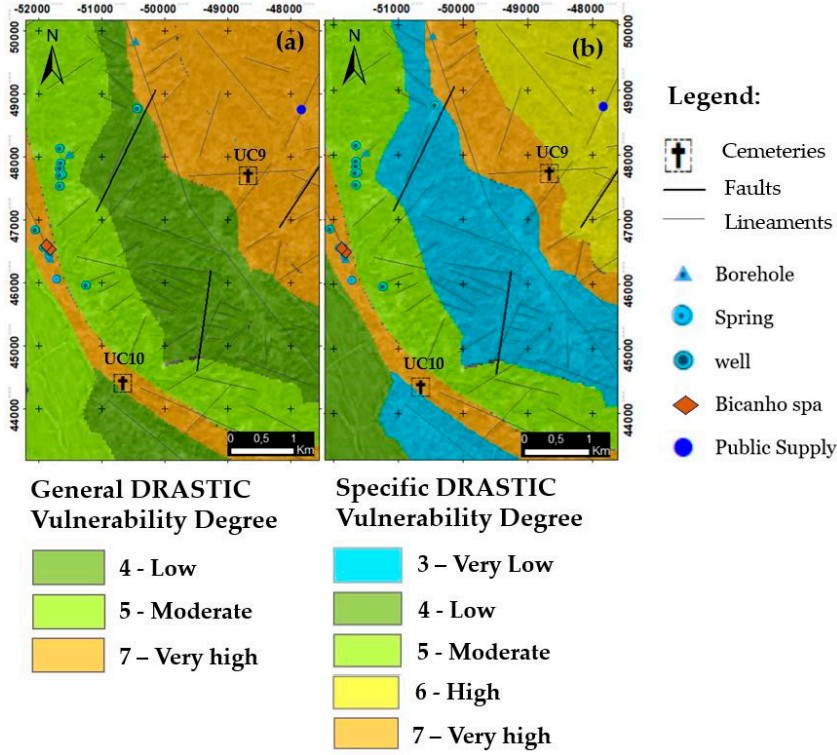

**Figure 12.** (**a**) Vulnerability map to pollution—General Drastic index in degrees; (**b**) Vulnerability map to pollution—Specific Drastic index in degrees.

### 3.3. Final Considerations

Despite the large number of studies that have evaluated the quality of groundwater under the influence of cemeteries [140–142], relatively few have looked at soil and unsaturated

zone characteristics in their evaluations. The only study that examined cemeteries was conducted by Razack and Sinan [73], who discovered that the observed DRASTIC indices ranged from 71 to 204. By applying the alternative GOD method to create vulnerability maps in four cemeteries in Santa Maria (Brazil), Kemerich et al. [143] determined that these cemeteries were the cause of bacterial contamination in the groundwater.

Owing to urban sprawl, a growing population, and ongoing conflicts between different land uses, the number of deaths is currently rising, while the amount of available land is decreasing. According to several sources [35,56,141], cemeteries should be located 250–500 m away from sources of potable groundwater and 30 m away from water courses or springs to reduce the risk of groundwater contamination; sands underlain by impermeable layers, for example, are not suitable as a burial substrate due to their high permeability; It is advantageous to have a thick aeration layer and a deep underground water table; the ground between graves and tombs must be made watertight; they must not be situated in sloped terrain or areas susceptible to landslides; there are no water-filled graves. According to previous studies, cemeteries have a high potential for pollution, especially if improperly built [144]. Cemeteries should have their surrounding groundwater and surface water quality investigated. In the absence of specific guidelines, monitoring should adhere to the Landfill Directive's best practices for water monitoring near landfill sites (Directive 1999/31/EC) [53].

Human decomposition can contaminate groundwater in the vicinity of cemeteries, but not because of any specific toxicity, but rather because it raises naturally occurring organic and inorganic substance levels to a point where the groundwater becomes unsuitable for any use [145,146]. Cemetery and burial ground risk management has been researched [28,127,147–152]. Increased nutrient concentrations, particularly nitrate compounds [7,13], have been found, and groundwater has been identified as the primary cemetery pollutant receptor [28,144,153–155].

The impact of numerous anthropogenic sources of pollution is the driving force behind most studies on groundwater vulnerability assessment; however, because of the location of the equipment and the lack of other notable nearby sources of pollution, this study particularly focused on pollution from cemeteries. People who use contaminated groundwater as their household water supply are at risk of spreading regional epidemics. As a result, because it contains important information, management organisations for cemeteries as well as entities in charge of environmental vulnerability and public health vigilance should replicate this study. A risk-based decision-making framework proposed by Pollard et al. [155] has been widely adopted in the UK and other European countries.

Future societal challenges will encourage the construction of technologically advanced cemeteries with digital systems (humidity, temperature, pH, and physical–chemical parameters sensors) that will allow the state of degradation of bodies to be accurately assessed without the need to open the graves. Currently, "green funeral" practices are increasing, where a tree is planted next to the buried body, and there are already forest cemeteries, eco-cemeteries, and natural memorial reserves. In the future, new contaminants will emerge related to the development of medical, industrial, and agricultural practices (so-called emerging contaminants), which will generate concern in the management of municipal services, such as cemeteries, as they can affect the water cycle.

## 4. Conclusions

This investigation made it possible to identify areas at risk of groundwater contamination from surface runoff from two cemeteries in the Soure region (Portugal), through the construction of a vulnerability map based on the DRASTIC and DRASTIC-specific indices and applying GIS tools and operations. Cemeteries can be a significant source of water contamination, particularly in vulnerable areas where this practice is the main source of pollution. The vulnerability map allowed for the identification of areas with different susceptibilities to contamination (ranging from "Low" to "Very high" for the DRASTIC index and from "Very Low" to "Very high" for the Specific DRASTIC).

Both cemeteries are in an area of high vulnerability to aquifer contamination, though UC10 is slightly more vulnerable in quantitative terms. Its location in an area with a lower slope (2%), which promotes infiltration, along with a higher drainage density and more favourable soil occupation and use—along with a higher TWI—all contribute to this. UC9 is in an area that has higher line density, SPI value, distance to rivers, and NDVI, but the environment is not as favourable for aquifer recharge and infiltration as UC10. The two cemeteries are situated in nearly identical lithological units in terms of hydraulics, which justifies the same vulnerability in terms of quality. Because the UC10 cemetery is close to the mineral resource of the Bicanho Medical Spa, within the same aquifer unit, and is a highly unique and sensitive resource, it must be closely monitored.

Hydrogeological cartography and groundwater vulnerability maps are excellent resources for helping the description, analysis, modelling, and communication of groundwater resource management. The production of maps from hydrogeological models like the DRASTIC index is made possible by the high potential of GIS for processing and analysing complex geo-referenced data. Particularly now that space is an issue in densely populated areas, GIS has shown to be an effective cemetery management tool.

**Author Contributions:** Conceptualization, V.G., A.A., L.F.G. and V.C.; methodology, V.G., A.A. and L.F.G.; software, V.G. and P.G.A.; validation, A.A., P.G.A., L.F.G. and V.C.; formal analysis, V.G., A.A. and L.F.G.; investigation, V.G., A.A. and L.F.G.; resources, V.C.; data curation, V.G.; writing—original draft preparation, V.G.; writing—review and editing, A.A., P.G.A., L.F.G. and V.C.; visualization, V.G.; supervision, A.A., L.F.G. and V.C.; project administration, V.C.; All authors have read and agreed to the published version of the manuscript.

**Funding:** This research received no external funding.

**Data Availability Statement:** The data presented in the manuscript were produced by us, either by applying the ArcGIS 10.8.2 or by personalised calculation based on the mathematical expressions presented. These data have not been published.

**Acknowledgments:** The authors are very grateful for the support granted by the Research Unit GeoBioTec, through the project reference UIDB/04035/2020, funded by the Fundação para a Ciência e a Tecnologia, IP/MCTES through national funds (PIDDAC).

**Conflicts of Interest:** The authors declare no conflict of interest.

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
