# Peer review of "Delineation of Potential Groundwater Zones and Assessment of Their Vulnerability to Pollution from Cemeteries Using GIS and AHP Approaches Based on the DRASTIC Index and Specific DRASTIC"

_water, doi:10.3390/w16040585_

Round 1

Reviewer 1 Report

Comments and Suggestions for Authors

The paper discusses the assessment of groundwater potential and vulnerability to contamination, focusing on factors such as lithological units, slope, drainage density, and soil. It highlights concerns about groundwater contamination by leachate, particularly in areas near cemeteries. The study utilizes methods such as the Analytical Hierarchy Process (AHP) and Remote Sensing (RS) to identify groundwater potential zones and assess vulnerability. Various studies on groundwater vulnerability and potential zones using different methods and technologies are also reviewed. 

1.Line 44-45 and line 63,64 are similar lines. 

2.Line 106-107 they were talking about several indicators. Again, in line 108-112 similar techniques are talked about. The redundant sentences can be removed. 

3.Line 113 What are the other techniques being referred to here in the sentence? Either mention the techniques or remove the sentence from the text altogether.  

4.Line 122 (modified DRATICL model) There is no DRATICL model. The authors must check the manuscript for such typos.  

5.Line 123-127 Results from other studies do not constitute the author’s own view. The authors can discuss techniques employed in other studies or text in line with the objectives of the authors, which is to assess the pollution from cemeteries.  

6.Line 152 other methods also fit well with GIS framework and give good result like GOD, SINTACS, EPIK, AVI. The authors must state specifically their reason for using DRASTIC over other techniques?  

7. Line 238  Figure 2 mentions the various datasets used in the study. A detailed table should be put in the text to explain data, data sources and the resolution. What kind of rainfall data is used in the study is not mentioned anywhere which makes it confusing for the reviewer to fully appreciate the datasets used. Also why is landsat 8 image used instead of sentinel-2 which has a better spatial resolution.  

8 .Line 420 How is the Ground Water Potential Zones map linked to DRASTIC model? This important information should be included in the text.  

9.Table 9 what is the basis of deciding the ranges of table?                 

10. The authors should keep either table 2 or table 4.  

11. The authors should make the introduction concise and up to the point. Unnecessary text should be excluded.

Comments on the Quality of English Language

A lot of repetitions were found. Proofreading is required. 

Author Response

Dear reviewer,

Thank you very much for the suggestions and corrections that allowed us to improve the quality of the manuscript.

1. Line 44-45 and line 63,64 are similar lines.

The sentence in lines 44–45 was rewritten to avoid repetition: "People and society have long considered contaminated groundwater near established and unplanned cemeteries to be an urgent concern [7], because it is a slow, chronic, and asymptomatic process [8,9], and should be referred to as decomposition labs [10]."

2. Line 106-107 they were talking about several indicators. Again, in line 108-112 similar techniques are talked about. The redundant sentences can be removed.

The sentences were rewritten to avoid repetition: “Because groundwater vulnerability cannot be measured directly [43], several indicators have been proposed to assess current groundwater quality or predict future scenarios [44,45]. Taghavi et al. [46] classified these evaluation methods into four categories: (i) overlay- and index-based methods [40,42,43], (ii) process-based simulation models [43,47], (iii) statistical methods (including orthodox and Bayesian methods) [42,43,48,49], and (iv) hybrid methods [50,51].”

3. Line 113 What are the other techniques being referred to here in the sentence? Either mention the techniques or remove the sentence from the text altogether.

The suggestion was included, and the sentences were written: “Other techniques have been put forth to assess the vulnerability of groundwater resources. These include the model of intrinsic groundwater vulnerability and specific vulnerability to pesticide pollution [52,53], techniques for determining karst aquifer vulnerability [54], an approach that incorporates impact modelling, and an index-based approach to determine how vulnerable groundwater resources are to climate change [55].”

4. Line 122 (modified DRATICL model) There is no DRATICL model. The authors must check the manuscript for such typos.

It was a mistake. It was corrected in all text.

5. Line 123-127 Results from other studies do not constitute the author’s own view. The authors can discuss techniques employed in other studies or text in line with the objectives of the authors, which is to assess the pollution from cemeteries.

The suggestion was included, and the sentences were written:The results indicated that landfills and industries were the most likely sources of pollution in the study area, and high vulnerability was observed in regions with shallow groundwater depths and high net recharge.”

The following sentence was removed: These areas are associated with high aquifer recharge rates, shallow water table depths, and highly permeable aquifer materials.

6. Line 152 other methods also fit well with GIS framework and give good result like GOD, SINTACS, EPIK, AVI. The authors must state specifically their reason for using DRASTIC over other techniques?

"The suggestion was included, and the sentences were written: “Furthermore, because the Drastic Index doesn't require complex numerical analysis or a multi-parameter simulation process, it is a model that uses less processing power. More significantly, though, it has a low application cost and produces excellent results. Because so many input data points are used, the DRASTIC index improves evaluation performance, which helps reduce the impact of errors on the final product. “

7. Line 238 Figure 2 mentions the various datasets used in the study. A detailed table should be put in the text to explain data, data sources and the resolution. What kind of rainfall data is used in the study is not mentioned anywhere which makes it confusing for the reviewer to fully appreciate the datasets used. Also why is landsat 8 image used instead of sentinel-2 which has a better spatial resolution.

The suggestion was included, and the sentences were modified: “Although the spatial resolution of the satellite images provided by ESA (European Space Agency) Sentinel-2 is generally higher, this implies more clarity and detail but also more data and storage. The Landsat-8 satellite is distinguished by the presence of thermal bands as well as band-8 (panchromatic), which is useful for improving image spectral resolution, and data are distinguished by a high radiometric resolution (16 bits), allowing the measurement of subtle variations in surface conditions.”

The following table has been added:

Table 1. Data used for creating GWPZs input data.

Data Type

Source

Format

Cell size

Date

Used to Produce

DEM

USGS

Raster

30x30 m

2022

Lineament density, NDVI, DTM - Distance to Rivers, TWI, Slope, SPI, Drainage density

Rainfall

SNIAMB

Shapefile polygon (1: 1, 000, 000) converted to raster

1931-1960

Annual precipitation - Recharge

Geology

LNEG

Shapefile polygon (1:500,000) converted to raster

1992

Geology

LULC

DGT

Shapefile polygon (1:25,000) converted to raster

2018

LULC

DEM - Digital Elevation Model; USGS - United States Geological Survey; DTM - Digital Terrain Model; SNIAMB – ‘Sistema Nacional de Informação de Ambiente’; LNEG – ‘Laboratório Nacional de Energia e Geologia’; LULC – Land use/Land cover; DGT – Direção Geral do Território.

The following sentence was added: Although more recent total precipitation data were calculated at the study site, data from 1931-1960 were used in the GIS environment because they were available in polygon shapefile format and the most recent data were contained within the polygon.

8. Line 420 How is the Ground Water Potential Zones map linked to DRASTIC model? This important information should be included in the text.

The following sentence was added: The cartography created for the GWPZs will be used to map the areas where aquifer recharge is favoured, as well as to define the various indices in the R parameter used in the DRASTIC index.

9. Table 9 what is the basis of deciding the ranges of table?

The Final Table was created by Reference [130]'s author; we used it as it was published in reference [130] because we recognised it as accurate and appropriate for application to our data. DRASTIC index (ID) ranges presented in the Table's first column on the left are the same suggested by the authors of the first publication on DRASTIC index [40]. Furthermore, as can be seen in the table below, we also established a parallel column of colours in addition to those classes.

The purpose of selecting these colours was to facilitate the cartographic representation of the "final message" regarding the degrees of pollution vulnerability. Areas with the greatest potential for issues are indicated by warm colours (red, orange, and yellow), whereas areas with less vulnerability to groundwater pollution are indicated by cool colours (blue, indigo, and violet). Two different green hues define the intermediate bands. According to reference [132], colour notes are significant because, as previously mentioned, the other columns in Table 10 are the outcome of didactically communicating the message to any user. As a result, the final column, qualitative representation, is far more informative for the average reader than the numeric column that only contains the ID value. It is also mentioned that the ID varies between 23 and 226; ID = 23 corresponds to PV = 0% and ID = 226 to PV = 100%. This variation in ID served as the foundation for the creation of the classes based on "potential vulnerability" (PV).

10. The authors should keep either table 2 or table 4.

Both tables must be maintained for the study to be complete and concise, as one is the next step of the other in the applied methodology, and the second is the result of the first's "normalisation". Keeping only one of them makes no sense.

11. The authors should make the introduction concise and up to the point. Unnecessary text should be excluded.

 The introduction has been revised and repetitive text has been merged or deleted.

Reviewer 2 Report

Comments and Suggestions for Authors

In this article, the authors evaluated the groundwater contamination risk around areas with cemeteries using various software and models such as GIS and DRASTIC index. The paper is well written and various information is given (laws, models etc.). The modelling procedure is adequately described. The results are discussed in detail and various parameters were taken into account. Prior to publication I suggest to address the following minor issues:

·        Please give more information about the subfigures in Figure 1.

·        Please provide briefly the rational for the class rank in Table 1. For example, why for the rainfall the spacings/categories are not equal e.g. the first is 0-298 mm/year and the second spacing is 298-740. In addition, these values for rainfall (from data of years 1931-1960), are representative for Portugal today? In line 618 it is mentioned that the annual precipitation is 852,4 mm.

·        Figure 7. In the legend it is mentioned a and b but there is only one figure.

·        Line 49 releases not release

·        Line 122 DRATICL or DRASTIC?

Comments on the Quality of English Language

minor editing is required, for example:

·        Line 49 releases not release

·        Line 122 DRATICL or DRASTIC?

Author Response

Dear reviewer,
Thank you very much for the suggestions and corrections that allowed us to improve the quality of the manuscript.

1. “Please give more information about the subfigures in Figure 1”.

A new image was inserted.

2. “Please provide briefly the rational for the class rank in Table 1. For example, why for the rainfall the spacings/categories are not equal e.g. the first is 0-298 mm/year and the second spacing is 298 740. In addition, these values for rainfall (from data of years 1931-1960), are representative for Portugal today? In line 618 it is mentioned that the annual precipitation is 852,4 mm”.

When we classified the data, we could have used one of the many standard classification methods included in ArcGIS Pro, or we could have created a custom class ranges.

Natural Breaks Jenks classifies data based on natural groupings. Class breaks are designed to align similar values and highlight differences between classes. The features are divided into classes, with boundaries set where there are significant differences in data values.

Natural breaks are data-specific classifications that are ineffective for comparing multiple maps constructed from disparate underlying information.

Rainfall performs better in classification using the Jenks Natural Breaks algorithm.

The following sentence has been added: “Although more recent total precipitation data were calculated at the study site, data from 1931-1960 were used in the GIS environment because they were available in polygon shapefile format and the most recent data were contained within the polygon.”

3. Figure 7. In the legend it is mentioned a and b but there is only one figure.

The following new figure has been inserted.

  1. Line 49 releases not release.

    It was corrected.

  2. Line 122 DRATICL or DRASTIC?  This paragraph was deleted.

Round 2

Reviewer 1 Report

Comments and Suggestions for Authors

The authors have satisfactorily responded to the raised queries.

Comments on the Quality of English Language

Okay.